# Clipping Makes Distributed and Federated Asynchronous SGD Robust to Stragglers

**Samuel Erickson** [1]   **Mikael Johansson** [1]

## Abstract

In modern machine learning, parallelization of training is an important strategy for increasing scale. Asynchronous stochastic gradient descent (ASGD), which maximizes the utilization of available hardware by avoiding waiting for slow workers. However, with constant step sizes, the convergence of ASGD is nonetheless affected negatively by slow workers due to large delays in updates. At the same time, it has been empirically observed in asynchronous training of deep learning models that gradient clipping "stabilizes" training. In this work, we provide a theoretical justification for this behavior, as we show that clipping removes the dependence of the maximum delay in the oracle complexity. We employ a sub-Weibull model of gradient noise which generalizes sub-Gaussian and sub-exponential distributions to more heavy-tailed distributions, motivated by empirical observations in deep learning. We show convergence in expectation, and the first time in asynchronous optimization, convergence with high probability.

## 1. Introduction

In recent years, parallelism has become the primary strategy to accommodate the increasing scale of machine learning models and the datasets used to train them. The classical parallel distributed optimization algorithm for general machine learning is Minibatch SGD. However, synchronization in Minibatch SGD means model updates cannot proceed until all workers have finished, leading to faster workers idling. This means that iterations are only as fast as the *slowest* worker. If computation times are uniform across workers, this is not a problem, but in many real-world settings this is far from being the case. For instance, these straggler

workers are present when there is heterogeneous hardware or network latency. In some settings, workers may even drop out entirely due to network issues.

To increase worker utilization and decrease idling, it is natural to consider removing locking, resulting in asynchronous SGD (ASGD) (Nedić et al., 2001; Tsitsiklis et al., 2003; Feyzmahdavian et al., 2016; Nguyen et al., 2018). Introducing asynchrony can substantially increase gradient throughput compared to Minibatch SGD, but it does introduce a new challenge: stale gradients. That is, instead of slowing down per-iteration run-time, stragglers slow down convergence due to the noise in their updates that arise from large delays in gradient computations. Both empirically and theoretically, the convergence of "vanilla" ASGD with constant step size is slowed down by a large maximum delay (Koloskova et al., 2022; Wu et al., 2022). For this reason, several works have proposed variants of ASGD where the contributions from workers with large delays are de-emphasized, to obtain convergence rates that are independent of the maximum delay. However, it was proved by Mishchenko et al. (2022) and Koloskova et al. (2022) that vanilla ASGD is in fact independent of the maximum delay when the gradients are globally bounded. This naturally raises the following question:

*Does gradient clipping remove the effect of the maximum delay, making Clipped ASGD robust to stragglers?*

If clipping can be shown to provide robustness against the effect of stragglers, it may explain some previously unexplained empirical behavior. Chen et al. (2016) found that in training large deep learning models, gradient clipping was necessary to "stabilize" asynchronous training, but was not necessary for synchronous training. For heterogeneous optimization (*e.g.*, federated learning) this may also be particularly significant. Since delay-adaptive strategies are biased toward faster workers, they do not converge to a stationary point in the heterogeneous data case. Can we use clipping to make training robust to stragglers without introducing a bias?

For these reasons, the focus of this work is to provide convergence guarantees for Clipped ASGD that are robust to stragglers. Moreover, in the asynchronous setting, high probability guarantees are especially interesting, as guaran-

[1]School of EECS, KTH Royal Institute of Technology, Stockholm, Sweden. Correspondence to: Samuel Erickson <samuelea@kth.se>.

*Proceedings of the 43$^{rd}$ International Conference on Machine Learning*, Seoul, South Korea. PMLR 306, 2026. Copyright 2026 by the author(s).

tees in expectation do not capture the behavior of a single or few runs. Doing several runs is of course antithetical to the point of introducing asynchrony, which is to more effectively utilize available hardware. Despite this, there are to the best of our knowledge no prior results showing high probability convergence under asynchrony. Therefore, we focus on providing guarantees with high probability in addition to guarantees in expectation.

**Contributions.** The main contributions of this work are two-fold and are as follows:

- We show that Clipped ASGD is robust to stragglers under a gradient noise model which allows us to model the heavy tails seen in deep learning. Specifically, we obtain rates that do not depend on the maximum delay in both the *homogeneous* and *heterogeneous* cases. For the heterogeneous case, this is to the best of our knowledge the first asynchronous optimization algorithm that is independent of the maximum delay, suggesting that clipping is beneficial in asynchronous federated learning where severe stragglers are often present.

- We show that Clipped ASGD converges with high probability with a polylogarithmic dependence on the failure probability, where the degree depends on the tail parameter of the gradient noise. This is to the best of our knowledge the first result showing high probability convergence of an asynchronous optimization algorithm.

## 2. Related work

**Asynchronous optimization.** Asynchronous optimization for machine learning has gained much interest since the works of Agarwal & Duchi (2011), Recht et al. (2011) and Dean et al. (2012) due to the increasing size of models and the datasets they are trained on. However, asynchrony in optimization is not a new idea (*cf.* Bertsekas & Tsitsiklis, 1989). Asynchronous gradient descent and SGD date back to at least the works of Nedić et al. (2001) and Tsitsiklis et al. (2003), respectively.

Feyzmahdavian et al. (2016) proved an oracle complexity of $O(\sigma^2/\varepsilon^2 + \tau_{\max}^2/\varepsilon)$ for smooth convex objectives, where $\tau_{\max}$ is the maximum delay. Mania et al. (2017) later introduced perturbed iterate analysis for studying the convergence of asynchronous stochastic optimization, which Stich & Karimireddy (2020) adopted in order to improve the complexity. They showed a $O(\sigma^2/\varepsilon^2 + \tau_{\max}/\varepsilon)$ complexity for smooth convex objectives, and a $O(\sigma^2/\varepsilon^4 + \tau_{\max}/\varepsilon^2)$ complexity for smooth non-convex objectives. Similarly, Koloskova et al. (2022) used perturbed iterate analysis but on another virtual sequence to further improve the complexity. They show that ASGD with constant step size achieves

$O(\sigma^2/\varepsilon^4 + \sqrt{\tau_{\max}\tau_C}/\varepsilon^2)$ complexity in the smooth non-convex case, where $\tau_C$ is the number of active workers, *i.e.*, the concurrency.

To mitigate the adverse effects of stragglers, several works have explored delay-adaptive optimization algorithms (McMahan & Streeter, 2014; Sra et al., 2016; Zhang et al., 2016; Zheng et al., 2017; Hannah & Yin, 2018; Cohen et al., 2021; Mishchenko et al., 2022; Koloskova et al., 2022; Wu et al., 2022). Picky SGD, proposed by Cohen et al. (2021), was the first ASGD variant to achieve the complexity $O(\sigma^2/\varepsilon^4 + \tau_C/\varepsilon^2)$, completely removing the dependence on the maximum delay. They achieved this by discarding gradients that are excessively stale. Mishchenko et al. (2022); Koloskova et al. (2022); Wu et al. (2022) later showed that it is also possible to achieve this by adapting the step size online using the realized delays. Extending the theory to incorporate time complexity, (Maranjyan et al., 2025) showed that by dropping gradients that exceed a certain delay, it is possible to achieve the optimal time complexity.

Another recent development is asynchronous federated learning (FL). Due to the heterogeneity in hardware capabilities, network latencies, and size of local datasets among workers, severe stragglers are common in cross-device FL, making asynchrony very attractive. It has been shown in large production FL deployments that asynchronous training can lead to substantial speed-ups and reductions in communication overhead (Huba et al., 2022). An early asynchronous FL method is FedAsync (Xie et al., 2020), which was shown to outperform synchronous FL when the maximum staleness was small. Koloskova et al. (2022) also studied a federated variant of ASGD, showing a similar oracle complexity as vanilla ASGD in the homogeneous case. In order to make asynchronous training compatible with privacy mechanisms, Nguyen et al. (2022) propose FedBuff which uses buffered aggregation. Wang et al. (2023) take a control variate approach with their method CA$^2$FL to reduce the effect of data heterogeneity. Recently, (Maranjyan & Richtárik, 2026) proposed Ringleader ASGD, which achieves the optimal time complexity under data heterogeneity.

In Table 1, we summarize the oracle complexities of related works and ours. For current surveys on asynchronous and parallel optimization, see Ben-Nun & Hoefler (2019) and Feyzmahdavian & Johansson (2023).

**Gradient clipping.** Gradient clipping is a widely used heuristic for dealing with instability in optimization, with Alber et al. (1998) being an early work making use of this technique. Mai & Johansson (2021) showed that gradient clipping can significantly improve the stability of SGD for non-smooth convex functions with rapidly growing sub-gradients. For non-convex learning, Mikolov et al.

*Table 1.* Oracle complexities (up to polylogarithmic factors) for homogeneous and heterogeneous smooth non-convex optimization. Here $\tau_C$ is the concurrency and $\tau_{\max}$ is the maximum delay, which is always larger than $\tau_C$.

| Algorithm | Homogeneous | Heterogeneous |
|---|---|---|
| Vanilla ASGD (Koloskova et al., 2022) | $\frac{\sigma^2}{\varepsilon^4} + \frac{\sqrt{\tau_{\max}\tau_C}}{\varepsilon^2}$ | $\frac{\sigma^2+\zeta^2}{\varepsilon^4} + \frac{\zeta\tau_C}{\varepsilon^3} + \frac{\sqrt{\tau_{\max}\tau_C}}{\varepsilon^2}$ |
| Delay-adaptive ASGD (Cohen et al. 2021; Koloskova et al. 2022; Mishchenko et al. 2022) | $\frac{\sigma^2}{\varepsilon^4} + \frac{\tau_C}{\varepsilon^2}$ | N/A |
| FedBuff (Nguyen et al. 2022; Wang et al. 2023) | N/A | $\frac{\sigma^2+\zeta^2}{\varepsilon^4} + \frac{\tau_{\max}\tau_C}{\varepsilon^2}$ |
| Clipped ASGD (This work) | $\frac{\sigma^2}{\varepsilon^4} + \frac{\sigma\tau_C}{\varepsilon^3} + \frac{\tau_C}{\varepsilon^2}$ | $\frac{\sigma^2+\zeta^2}{\varepsilon^4} + \frac{(\sigma+\zeta)\tau_C}{\varepsilon^3} + \frac{\tau_C}{\varepsilon^2}$ |

(2012) and Pascanu et al. (2013) proposed gradient clipping for dealing with the so-called exploding gradient problem. Later, Zhang et al. (2020b) provided a theoretical justification for using clipping in this context. By generalizing the standard smoothness assumption to $(L_0, L_1)$-smoothness, they showed that gradient clipping may enable the use of significantly larger step sizes. Zhang et al. (2020a) then improved upon the dependence on problem-specific parameters and allowed for momentum. However, Zhang et al. (2020a;b) both used the strong uniformly bounded gradient noise model. Relaxing to the standard bounded variance model, Koloskova et al. (2023) showed that Clipped SGD may still enjoy large step sizes, but has a relatively large *irreducible* error term.

Another important role that gradient clipping plays is in dealing with heavy-tailed gradient noise. Several works have showed that the gradient noise in the training of deep learning models have much heavier tails than sub-Gaussian distributions model (Simsekli et al., 2019; Panigrahi et al., 2019; Gurbuzbalaban et al., 2021). As a response, subsequent works have concerned SGD and its variants under different models of heavy-tailed noise.

Zhang et al. (2020c) showed that Clipped SGD is convergent in expectation for noise with bounded $q$-moment for $q \in (1, 2]$, whereas vanilla SGD can diverge for $q < 2$. Later, Cutkosky & Mehta (2021) and Nguyen et al. (2023) showed high probability convergence of Clipped SGD under the same noise model. In addition, Cutkosky & Mehta (2021) showed that SGD fails to attain a logarithmic dependence on the failure probability even in the bounded variance setting $q = 2$. More recently, Hübler et al. (2025) connected gradient clipping to normalization and established improved, parameter-free convergence guarantees for non-convex optimization under heavy-tailed noise with only bounded $p$-th moments. They further proved tight sample complexity

bounds and high-probability convergence. Taking another approach to modeling heavy tails, Li & Liu (2022) and Madden et al. (2024) adopted a sub-Weibull (Vladimirova et al., 2020) noise assumption, and showed high probability convergence of Clipped SGD.

In privacy-preserving machine learning, clipping also plays a structural role rather than a purely optimization-oriented one. Early work on differentially private empirical risk minimization and stochastic optimization explicitly relies on globally bounded gradients to calibrate additive noise mechanisms (Bassily et al., 2014; Dwork et al., 2014), and this requirement was operationalized in deep learning through gradient clipping in differentially private SGD (Abadi et al., 2016). Chen et al. (2020) later studied the bias that clipping introduces in differentially private SGD through a geometric lens. Extensions to federated learning further have emphasized clipping as a prerequisite for aggregating privatized updates across clients (McMahan et al., 2018).

## 3. Problem and computational setup

We consider the unconstrained optimization problem

$$\text{minimize} \quad f(x) = \sum_{i=1}^n \mathbf{E}_{\xi \sim \mathcal{D}_i}[F_i(x, \xi)] \qquad (1)$$

where $F_i : \mathbf{R}^d \times \mathcal{S} \to \mathbf{R}$ for a sample space $\mathcal{S}$. We define $f_i(x) = \mathbf{E}_{\xi \sim \mathcal{D}_i}[F_i(x, \xi)]$. We denote the standard inner product $\langle \cdot, \cdot \rangle$ and the Euclidean norm $\| \cdot \|$. We make the standard smoothness assumption on the objective $f$:

**Assumption 3.1.** The objective function $f$ is $L$-smooth, meaning $\|\nabla f(x) - \nabla f(y)\| \leq L\|x - y\|$ for every $x, y \in \mathbf{R}^d$. Furthermore, $f$ is bounded from below by $f^\star > -\infty$.

The *clipping operator* $\text{clip}_c : \mathbf{R}^d \to \mathbf{R}^d$ with *clipping ra-*

*dius* $c > 0$ is defined by

$$\mathbf{clip}_c(x) = \min\left\{1, \frac{c}{\|x\|}\right\} x$$

for $x \in \mathbf{R}^d \setminus \{0\}$ and $\mathbf{clip}_c(0) = 0$. Note that clipping $x$ is equivalent to projecting $x$ onto the ball $\{y : \|y\| \le c\}$.

In this setup, we assume $n$ workers are available for parallel computation, with each worker $i$ serving as a stochastic oracle that returns $g_t^i(x_t) = \mathbf{clip}_c(\nabla F_i(x_t, \xi_t^i))$ when queried with $x_t$, where $\xi_t^i \sim \mathcal{D}_i$. In a real-world system, these workers may be GPUs in a cluster, cores in a CPU, mobile devices, *etc*.

Crucial to most analyses of gradient clipping is controlling the error of the clipped stochastic gradient when the full gradient is small (relative to the clipping threshold). This generally involves controlling the tail of the gradient noise, *i.e.*, bounding the probability $\mathbf{P}(\|\nabla F(x, \xi) - \nabla f(x)\| > \alpha)$ with a function of $\alpha \ge 0$. The works of Zhang et al. (2020a;b) consider the uniformly bounded noise model, when the tail probability is zero for $\alpha \ge \sigma$ for some parameter $\sigma > 0$. While this technically holds true in typical empirical risk minimization, it requires a very large value of $\sigma$ in comparison to the ordinary bounded variance model. Zhang et al. (2020b) also note that the noise model can be relaxed to sub-Gaussian noise, however, this assumption is still strong. Recent works in deep learning have showed that the stochastic gradients typically exhibit more heavy-tailed noise than sub-Gaussian random variables models (Simsekli et al., 2019; Panigrahi et al., 2019; Gurbuzbalaban et al., 2021). Koloskova et al. (2023) consider the ordinary bounded variance model and use Markov's inequality to bound the tail. However, Markov's inequality gives a pessimistic bound, leading to a relatively large error term even for large choices of clipping radius. Koloskova et al. show that this error is irreducible in worst-case analysis. Additionally, the convergence results in these works are in expectation, which do not characterize the behavior of Clipped SGD in a single run. This motivates the need for a model of gradient noise that gives us better tools to bound the tails, but still allows us to model the heavy-tails seen in deep learning.

**Definition 3.1.** A random variable $X: \mathcal{S} \to \mathbf{R}$ is called *sub-Weibull* if there exists positive quantities $\sigma$ and $\theta$ such that

$$\mathbf{E}[\exp((|X|/\sigma)^{\frac{1}{\theta}})] \le 2.$$

We denote $X \sim \mathsf{subW}(\theta, \sigma)$.

The class of sub-Weibull random variables is proposed by Kuchibhotla & Chakrabortty (2022) and Vladimirova et al. (2020). Kuchibhotla & Chakrabortty analyze linear regression and covariance estimation under this model, while Vladimirova et al. analyze the induced prior distributions in Bayesian neural networks. Note that sub-Weibull random variables generalize sub-Gaussian and sub-exponential random variables, which are recovered with $\theta = 1/2$ and $\theta = 1$, respectively.

**Assumption 3.2.** The stochastic gradients are unbiased and have sub-Weibull noise, meaning $\mathbf{E}_{\xi \sim \mathcal{D}_i}[\nabla F_i(x, \xi)] = \nabla f_i(x)$ and

$$\|\nabla F_i(x, \xi) - \nabla f_i(x)\| \sim \mathsf{subW}(\theta, \sigma)$$

for every $x \in \mathbf{R}^d$.

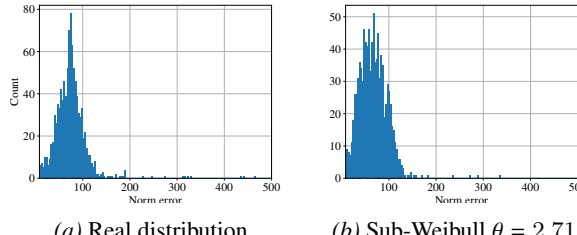

*(a)* Real distribution      *(b)* Sub-Weibull $\theta = 2.71$.

*Figure 1.* Histograms of (a) gradient errors in training of a ResNet-18 model on CIFAR-10 and (b) simulated sub-Weibull distribution. The empirical estimate of the tail parameter is $\theta = 2.71$.

This model of the gradient noise is used by Li & Liu (2022) and (Madden et al., 2024), who provide high probability guarantees for serial SGD. Note that if a random variable is sub-Weibull then it also has bounded second-moment. In particular, if $X \sim \mathsf{subW}(\theta, \sigma)$ then $\mathbf{E}|X|^2 \le C\sigma^2$, where $C$ only depends on $\theta$. Therefore, the quantity $\sigma^2$ is directly comparable to the one in the ordinary bounded variance assumption. The quantity $\theta$ on the other hand determines how heavy the tails are. This makes the class of sub-Weibull distributions ideal for modeling gradient noise in the training of modern machine learning models. In Figure 1, we have plotted the empirical norm gradient errors in training of a ResNet-18 model on the CIFAR-10 dataset, as well as a simulated sub-Weibull distribution using the estimate of the tail parameter $\theta$ obtained from the ResNet-18 training. We used the tail parameter estimate suggested by Vladimirova et al. (2020).

## 4. Homogeneous setting

We begin with the homogeneous special case of problem (1), where the objective functions $f_i$ are identical, *i.e.*, we have $F_1 = \dots = F_n$ and $\mathcal{D}_1 = \dots = \mathcal{D}_n$. This corresponds to the data centralized setup, where all workers have access to the full dataset. This setting gives us much freedom in how we utilize the workers, because we do not have to worry about introducing optimization bias from over-relying on the fast workers. Therefore, we consider Algorithm 1, where workers are at all times computing (or communicating) clipped

**Algorithm 1** Clipped ASGD (homogeneous setting)

**Input:** Initialization $x_0$, concurrency $\tau_C$, step size $\eta > 0$, clipping radius $c > 0$.

A subset $C_0$ of $\tau_C$ workers receive $x_0$ and start computing gradients

**for** $t = 0, \dots, T-1$ **do**

    Worker $i_t$ finishes computing $g_{t-\tau_t}^{i_t}(x_{t-\tau_t})$

    Server updates $x_{t+1} = x_t - \eta g_{t-\tau_t}^{i_t}(x_{t-\tau_t})$

    Server selects an inactive worker $j_t \in [n] \setminus C_t$ and updates the active set $C_{t+1} = (C_t \setminus \{i_t\}) \cup \{j_t\}$

    Worker $j_t$ receives $x_{t+1}$ and starts computing $g_{t+1}^{j_t}(x_{t+1})$

**end for**

---

gradients. This is the standard ASGD algorithm considered in *e.g.* (Agarwal & Duchi, 2011; Feyzmahdavian et al., 2016; Stich & Karimireddy, 2020; Mishchenko et al., 2022; Koloskova et al., 2022), differing only in the use of gradient clipping. Following Koloskova et al., we generalize to concurrency $\tau_C \le n$, although in the homogeneous case it is common to run with full concurrency. For instance, the server may prioritize workers that have returned gradients quickly in the past. Here, the server may select the next worker $j_t$ out of the inactive workers in any way, due to the homogeneity. Under maximum concurrency $\tau_C = n$ and the fixed computation model used by *e.g.* (Mishchenko et al., 2022; Tyurin & Richtárik, 2023), where each worker $i$ takes $h_i$ seconds to return a clipped gradient, Algorithm 1 takes at most $(\sum_{i=1}^n \frac{1}{h_i})^{-1}$ seconds per oracle call on average. Compare this to Minibatch SGD which takes $\frac{1}{n} \max_i h_i$ seconds per oracle call.

Central to recent analyses of ASGD is perturbed iterate analysis, where a virtual sequence is used to study the actual sequence $\{x_t\}_t$ of iterates (Mania et al., 2017; Stich & Karimireddy, 2020; Mishchenko et al., 2022; Koloskova et al., 2022). In the state-of-the-art analyses of ASGD (Mishchenko et al., 2022; Koloskova et al., 2022), it is shown that with a virtual sequence $\tilde{x}_t$ which evolves almost like serial SGD, the differences $\tilde{x}_t - x_t$ can be expressed as a sum of at most $n$ gradient steps. It is precisely here that, in our theoretical analysis, clipping turns out to be very beneficial for ASGD. It is in fact for the same reason the strong assumption of globally bounded gradients is beneficial in (Mishchenko et al., 2022; Koloskova et al., 2022). In particular, with initialization $x_0 = \tilde{x}_0$ and step size $\eta$, we define the virtual sequence by

$$\tilde{x}_1 = x_0 - \eta \sum_{i \in C_0} g_0^i(x_0), \quad \tilde{x}_{t+1} = \tilde{x}_t - \eta g_t^{i_t}(x_t) \quad (2)$$

and find that the differences $\tilde{x}_t - x_t$ can be controlled via the clipping radius $c$ and the step size $\eta$.

**Lemma 4.1.** *The sequence $\{x_t\}_t$ generated by Algorithm 1*

*and the sequence $\{\tilde{x}_t\}_t$ defined by* (2) *satisfy*

$$\|\tilde{x}_t - x_t\| \le \eta c \tau_C$$

*for all $t = 0, \dots, T-1$.*

Using this result, we may show that the convergence in expectation of Clipped ASGD does not depend on the maximum delay.

**Theorem 4.2.** *Suppose Assumptions 3.1 and 3.2 hold. Then there exists a constant step size $\eta$ and clipping radius $c$ such that for $\varepsilon \in (0,1)$, we have $\frac{1}{T} \sum_{t=0}^{T-1} \mathbf{E}\|\nabla f(x_t)\| \le \varepsilon$ within*

$$\widetilde{O}\left(\frac{\sigma^2}{\varepsilon^4} + \frac{\sigma\tau_C}{\varepsilon^3} + \frac{\tau_C}{\varepsilon^2}\right)$$

*iterations of Algorithm 1.*

Note that eventhough Clipped ASGD differs only in the middle term compared to the iteration complexity of Delay-adaptive ASGD, despite the algorithm not compensating directly for delays. In fact, if either $\tau_C = O(\sigma/\varepsilon)$ or $\sigma/\varepsilon = O(1)$, then Theorem 4.2 shows that Clipped ASGD achieves the same rate as up to polylogarithmic factors. Clipped ASGD is also favorable to Vanilla ASGD under globally bounded gradients $\|\nabla f(x)\| \le G$ or $G$-Lipschitz loss, which achieves the rate $O(\sigma^2/\varepsilon^4 + \tau_C G/\varepsilon^3 + \tau_C/\varepsilon^2)$, as $G$ will often be very large. Moreover, clipping also simplifies tuning. With Vanilla ASGD, the step size to achieve the rate $O(\sigma^2/\varepsilon^4 + \sqrt{\tau_{\max}\tau_C}/\varepsilon^2)$ critically depends on $\tau_{\max}$, which is generally unknowable a priori.

We can furthermore show convergence in high probability, which to the the best of our knowledge is the first time in asynchronous optimization.

**Theorem 4.3.** *Suppose Assumptions 3.1 and 3.2 hold. Then there exists a constant step size $\eta$ and clipping radius $c$ such that for $\varepsilon \in (0,1)$ and failure probability $\delta \in (0,1)$, we have $\mathbf{P}\left(\frac{1}{T} \sum_{t=0}^{T-1} \|\nabla f(x_t)\| \le \varepsilon\right) \ge 1 - \delta$ within*

$$\widetilde{O}\left(\frac{\sigma^2 \log^{2\theta}(1/\delta)}{\varepsilon^4} + \frac{\sigma\tau_C \log^{\theta}(1/\delta)}{\varepsilon^3} + \frac{\tau_C}{\varepsilon^2}\right)$$

*iterations of Algorithm 1.*

In particular, in the high probability analysis, we get an additional martingale difference term

$$Z_t = -\eta\langle \nabla f(\tilde{x}_t), g_t^{i_t}(x_t) - \mathbf{E}[g_t^{i_t}(x_t)]\rangle.$$

Gradient clipping lets us bound the norm of the arguments in the inner product, where Lemma 4.1 is necessary for the first argument. Subsequently applying Freedman's inequality (Lemma A.5) we find that with probability atleast $1 - \delta$,

$$\sum_{t=0}^{T-1} Z_t \lesssim \sum_{t=0}^{T-1} \left(\eta\|\nabla f(x_t)\|^2 + \eta^3 c^2 \tau_C^2 L^2\right)$$

$$+ \eta c^2 \log\frac{2}{\delta} + \sigma^2 \log^{2\theta}\frac{2T}{\delta}.$$

Theorem 4.3 shows that the rate differs from Theorem 4.2 only in a polylogarithmic dependence on the failure probability, with the degree being determined by the tail parameter $\theta$ from Assumption 3.2.

*Remark* 4.1. Since clipping does not affect the computation dynamics of ASGD, ignoring the small overhead of clipping the gradient, the same time complexity analysis applies. In the fixed computation model considered in (Tyurin & Richtárik, 2023; Maranjyan et al., 2025) we assume workers take at most $s_1 \leq ... \leq s_n$ seconds to compute gradients. Under this model, the time complexity of Clipped ASGD with full concurrency $\tau_C = n$ is obtained by multiplying the oracle complexity by the harmonic sum of computation times $(\sum_{i=1}^{n} \frac{1}{s_i})^{-1}$. If the computation times are known beforehand, then the concurrency and set of workers can be chosen to yield a time complexity of

$$\widetilde{O}\left(\min_{\tau_C}\left(\sum_{i=1}^{\tau_C}\frac{1}{s_i}\right)^{-1}\left(\frac{\sigma^2}{\varepsilon^4}+\frac{\sigma\tau_C}{\varepsilon^3}+\frac{\tau_C}{\varepsilon^2}\right)\right).$$

The same reasoning applies to the high probability analysis, since the worst-case computation dynamics in this model are deterministic.

## 5. Heterogeneous setting

We now turn to the general heterogeneous case of problem (1), in which every worker has access to its own data distribution and function $F_i$. In this setup, we have much less freedom in how we utilize workers, since over-relying on a few fast workers will bias towards their objectives. For this reason, standard ASGD does not in general converge in this setting (Mishchenko et al., 2022). Thus, we consider the same scheme for worker selection as for instance Koloskova et al. (2022) and Wang et al. (2023) in Algorithm 2.

---

**Algorithm 2** Clipped ASGD (heterogeneous setting)

**Input:** Initialization $x_0$, concurrency $\tau_C$ step size $\eta > 0$, clipping radius $c > 0$.
A subset $C_0$ of $\tau_C$ workers receive $x_0$ and start computing clipped gradients
**for** $t = 0, ..., T-1$ **do**
  Worker $i_t$ finishes computing $g_{t-\tau_t}^{i_t}(x_{t-\tau_t})$
  Server updates $x_{t+1} = x_t - \eta g_{t-\tau_t}^{i_t}(x_{t-\tau_t})$
  Worker $j_t \sim \mathsf{Uniform}\{1, ..., n\}$ receives $x_{t+1}$ and schedules $g_{t+1}^{j_t}(x_{t+1})$
**end for**

---

After a worker has finished computing a clipped gradient, a new worker is chosen uniformly at random among all workers, and is sent the updated model. This way, no bias is created toward faster workers. This does however potentially slow down the wall-clock time of the average oracle

call as compared to standard ASGD. Here, a worker can be sampled several times before finishing the first gradient computation, in which case a queue of gradients builds up on that worker.

We make the bounded first-order heterogeneity assumption on the functions $f_i$ which is commonly found in the federated learning literature.

**Assumption 5.1.** The functions $f_i$ have bounded heterogeneity, meaning $\|\nabla f_i(x) - \nabla f(x)\|^2 \leq \zeta^2$ for every $x \in \mathbf{R}^d$.

**Theorem 5.1.** *Suppose Assumptions 3.1, 3.2 and 5.1 hold. Then there exists a constant step size $\eta$ and clipping radius $c$ such that $\frac{1}{T}\sum_{t=0}^{T-1}\mathbf{E}\|\nabla f(x_t)\| \leq \varepsilon$ within*

$$\widetilde{O}\left(\frac{\sigma^2 + \zeta^2}{\varepsilon^4} + \frac{(\sigma+\zeta)\tau_C}{\varepsilon^3} + \frac{\tau_C}{\varepsilon^2}\right)$$

*iterations of Algorithm 2.*

Surprisingly, Theorem 5.1 shows that gradient clipping can remove the dependence on the maximum delay even in the heterogeneous case, where delay-adaptive schemes which do not in general converge. Compared this rate with the rate $O((\sigma^2+\zeta^2)/\varepsilon^4+\zeta\tau_C/\varepsilon^3+\sqrt{\tau_C\tau_{\max}}/\varepsilon^2)$ heterogeneous version of Vanilla ASGD, we have a significant improvement when delays are large. Moreover, for this result Koloskova et al. (2022) require an additional assumption that the average delay of a worker is independent from the number of times that worker is sampled, which may not be entirely realistic.

**Theorem 5.2.** *Suppose Assumptions 3.1, 3.2 and 5.1 hold. Then there exists a constant step size $\eta$ and clipping radius $c$ such that for $\varepsilon \in (0,1)$ and failure probability $\delta \in (0,1)$, we have $\mathbf{P}\left(\frac{1}{T}\sum_{t=0}^{T-1}\|\nabla f(x_t)\| \leq \varepsilon\right) \geq 1 - \delta$ within*

$$\widetilde{O}\left(\frac{(\sigma^2 + \zeta^2)\log^{2\theta}(1/\delta)}{\varepsilon^4} + \frac{(\sigma+\zeta)\tau_C\log^{\theta}(1/\delta)}{\varepsilon^3} + \frac{\tau_C}{\varepsilon^2}\right)$$

*iterations of Algorithm 2.*

This high-probability convergence guarantee may be particularly important in federated learning, as training runs in real-world deployments are typically done at most a few times because each run incurs substantial logistical, computational, and organizational cost. Coordinating participation across a large, dynamic population of client devices requires careful scheduling, incentives, and system overhead, and client availability cannot be reliably reproduced across runs. Moreover, federated training pipelines are tightly coupled to product release cycles, privacy reviews, and regulatory approvals, making repeated experimentation slow and expensive. As a result, retraining is not a cheap or repeatable process, making high-probability guarantees particularly relevant.

*Remark* 5.2. Quantifying the time complexity is more delicate in the heterogeneous case due to the sampling scheme required to preserve the unbiasedness of the stochastic gradients. For this reason, we have included experiments on the homogeneous and heterogeneous setting to empirically measure this slowdown.

# 6. Numerical experiments

We examine the effect of using gradient clipping in asynchronous optimization of neural network parameters. We conduct several experiments on the CIFAR-10 (Krizhevsky et al., 2009) and Shakespeare (Karpathy, 2015) datasets, in the homogeneous (shared memory) setting. To allow for different delay setups, we simulate asynchronous training with $n = 16$ workers, where half of them take 1 time unit to compute a gradient, and the other half takes $D \in \{4, 8\}$ time units. We use full concurrency $\tau_C = n$ in all runs. All experiments are averaged over three random seeds, and $2\sigma$ error bars are shown in the figures. The code used to run these experiments is available at https://github.com/samericks/clipped-asgd.

## 6.1. Homogeneous setting

We compare Clipped ASGD with three baseline methods in the homogeneous setting: Vanilla ASGD with constant step size, Delay-adaptive ASGD with step size rule proposed by Koloskova et al. (2022), and Ringleader ASGD (Maranjyan et al., 2025). For Clipped ASGD, we choose the best clipping radius $c \in \{2^k : k = -1, \dots, 2\}$, and for Ringleader ASGD we choose the best delay threshold $R \in \{2^k : k = 1, \dots, 4\}$. In all cases, the optimal hyperparameter value lies strictly within the search range.

**CIFAR-10.** We train a ResNet-18 model (He et al., 2016) on CIFAR-10. We sweep over $\eta \in \{2^{-9}, \dots, 2^{-1}\}$ and measure the simulated wall-clock time required to reach 80% test accuracy. We terminate runs that do not reach the target within 4,000 time units.

Figure 2 shows the results for delay factors $D \in \{4, 8\}$. Clipped ASGD consistently improves over Vanilla ASGD and Delay-adaptive ASGD across all delay settings. Here, Clipped ASGD reduces the minimum wall-clock time by $1.8\times$ relative to Vanilla ASGD, and by $1.5\times$ relative to Delay-adaptive and Ringmaster ASGD.

As predicted by the theory, increasing the delay primarily affects Vanilla ASGD, which requires substantially smaller step sizes to remain stable. While the best wall-clock time is not significantly worsened for Vanilla ASGD, it requires more fine-grained tuning to converge within the time budget. In contrast, the clipped and delay-adaptive methods remain robust to larger delays, with no significant shifts due to

larger delays.

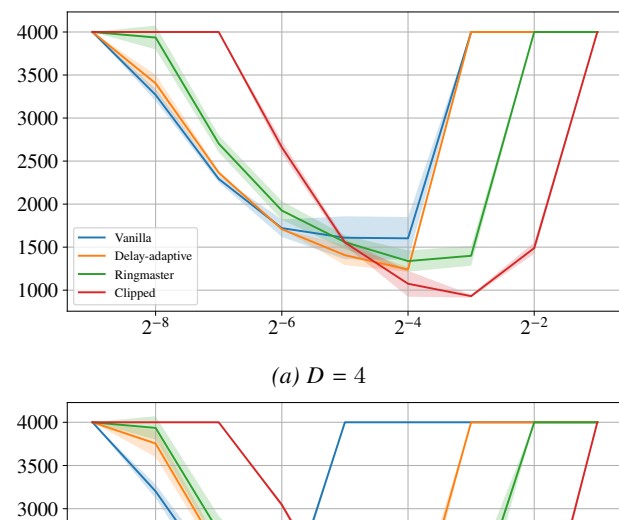

*(a) D = 4*

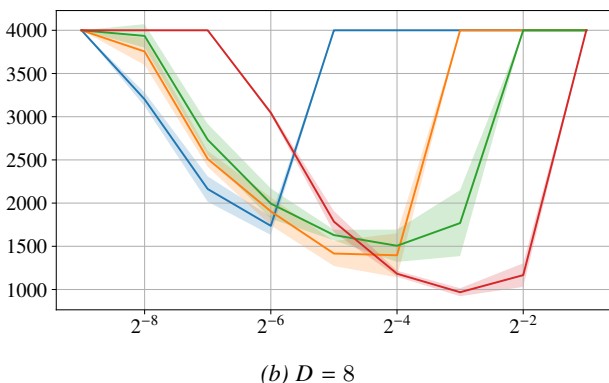

*(b) D = 8*

*Figure 2.* Simulated wall-clock time to reach 80% test accuracy on CIFAR-10 dataset with ResNet-18 architecture, when half of the 16 workers are $D$ times slower than the other half. The average time per oracle call here is 0.108 and 0.123 time units for $D = 4$ and $D = 8$, respectively.

**Shakespeare.** We next consider next-word prediction on the Shakespeare dataset using an LSTM architecture (Hochreiter & Schmidhuber, 1997) with dropout rate 0.2. We sweep over step sizes $\eta \in \{2^{-3}, \dots, 2^3\}$ and measure the simulated wall-clock time required to reach test perplexity 5.0. Runs are terminated after 4,000 time units.

The results are shown in Figure 3. Clipped ASGD again outperforms both Vanilla and Delay-adaptive ASGD. For $D = 4$, Clipped ASGD achieves the target perplexity $1.8\times$ faster than Vanilla ASGD, $2.1\times$ faster than Delay-adaptive ASGD, and $1.8\times$ faster than Ringmaster ASGD. For $D = 8$, Clipped ASGD is $2\times$ faster than Vanilla ASGD, $2.2\times$ faster than Delay-adaptive ASGD, and $1.4\times$ faster than Ringmaster ASGD.

## 6.2. Heterogeneous setting

We now consider asynchronous optimization under data heterogeneity. We compare Clipped ASGD against Vanilla ASGD and Ringleader ASGD (Maranjyan & Richtárik, 2026). Clipped and Vanilla ASGD use the uniform sam-

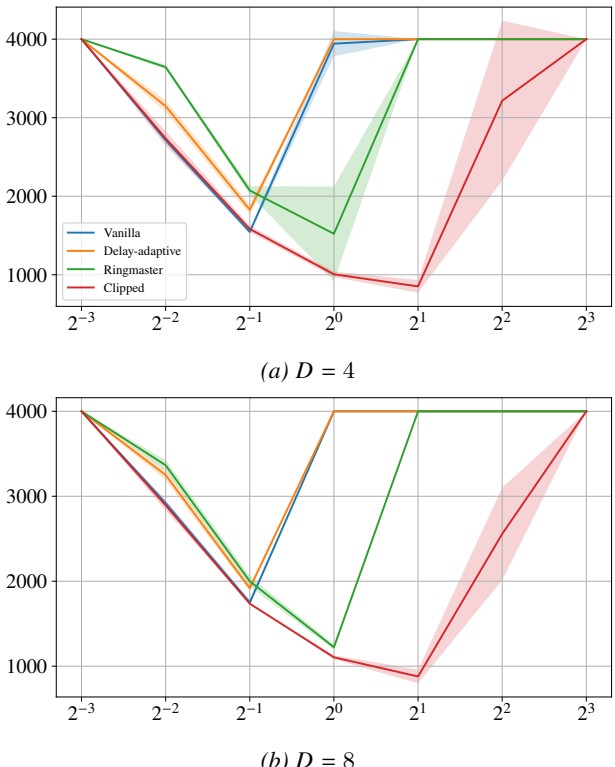

*(a) D = 4*

*(b) D = 8*

*Figure 3.* Simulated wall-clock time to reach test perplexity 5.0 on Shakespeare dataset with LSTM architecture, when half of the 16 workers are $D$ times slower than the other half. The average time per oracle call here is 0.108 and 0.123 time units for $D = 4$ and $D = 8$, respectively.

pling scheme described in Section 5. As before, we tune the clipping radius over $c \in \{2^k : k = -1, \dots, 2\}$.

Note however that in practical FL deployments, exact unbiasedness is often sacrificed in favor of faster process times. For example, many synchronous FL deployments aggregate updates once only a fraction (*e.g.*, 80%) of workers have responded, rather than waiting for the slowest participants. Thus, asynchrony can in fact *decrease bias*, since the slowest workers may participate at all (Huba et al., 2022).

**Label-skew CIFAR-10.** We revisit CIFAR-10, but now distribute the data heterogeneously across workers using a Dirichlet partitioning scheme. Following Nguyen et al. (2022); He et al. (2020); Diao et al. (2020), we sample client label distributions from a Dirichlet distribution with parameter $\alpha = 0.5$. We use a two-layer CNN architecture and sweep over step sizes $\eta \in \{2^{-9}, \dots, 2^{-1}\}$.

We measure the simulated wall-clock time required to reach 70% test accuracy, with a maximum runtime of 8,000 for $D = 4$ and 12,000 for $D = 8$. The results are shown in Figure 4. Clipped ASGD consistently improves over both baselines across heterogeneity levels. With delay factor

$D = 4$, clipping reduces the minimum wall-clock time by 1.2× relative to Vanilla ASGD and Ringleader ASGD. With $D = 8$, Clipped ASGD is 1.3× faster than Vanilla ASGD, and 1.2× faster than Ringleader ASGD. Note that due to the sampling schemes used to avoid oversampling the fast workers, the maximum delay is also effectively controlled at the expense of process time, which may explain why clipping yields a smaller improvement here than in the homogeneous experiments.

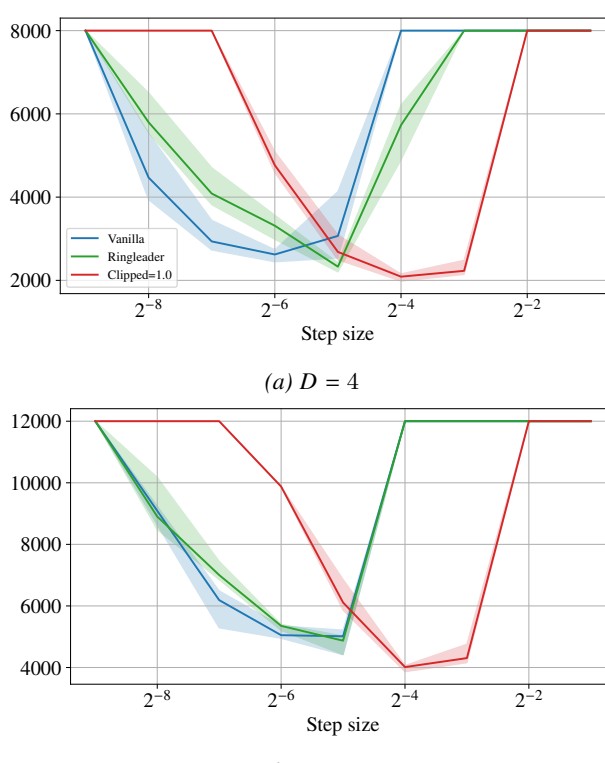

*(a) D = 4*

*(b) D = 8*

*Figure 4.* Simulated wall-clock time to reach test metric target on label skew CIFAR-10 dataset with a CNN architecture, when half of the 16 workers are $D$ times slower than the other half. The average time per oracle call is 0.337 and 0.668 time units for $D = 4$ and $D = 8$ respectively.

## 7. Conclusions and future work

This paper shows that gradient clipping fundamentally alters the effect of asynchrony in stochastic optimization. In particular, clipping removes dependence on the maximum delay in the convergence behavior of ASGD, yielding methods that are provably robust to stragglers. We provide convergence guarantees in both expectation and high probability, and show that these guarantees hold under homogeneous and heterogeneous settings. Empirically, we demonstrate that clipping consistently improves asynchronous training and can substantially outperform both Vanilla and Delay-adaptive ASGD across a range of architectures and delay regimes.

**Future work.** Our results suggest a broader connection between norm control and robustness to asynchrony, raising several directions for future investigation.

First, it is natural to ask how these effects extend beyond the standard smooth setting. In particular, under weaker assumptions such as $(L_0, L_1)$-smoothness, the interaction between gradient norms and staleness may become more pronounced, potentially making clipping even more beneficial.

Second, recent optimizers such as Muon (Jordan et al., 2024) and Scion (Pethick et al., 2025) achieve strong empirical performance by explicitly controlling update norms, through orthogonalization or constrained update steps. Since clipping provides a simple mechanism for norm control in asynchronous settings, it is natural to ask whether asynchronous variants of these methods inherit similar robustness to delays. We believe understanding this interaction between optimizer geometry and asynchrony is a promising direction for future work.

## Impact statement

This paper presents work whose goal is to advance the field of Machine Learning. There are many potential societal consequences of our work, none which we feel must be specifically highlighted here.

## Acknowledgement

This work was supported in part by the Knut and Alice Wallenberg Foundation through project KAW 2022.0050 and by the Wallenberg AI, Autonomous Systems and Software Program (WASP).

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

# A. Preliminaries

## A.1. Properties of Sub-Weibull random variables

**Theorem A.1** (Vladimirova et al. 2020)**.** *Let $X \colon S \to \mathbf{R}$ be a random variable. Then the following are equivalent:*

(i) *There exists a $K_1 > 0$ such that $\mathbf{P}(|X| \geq x) \leq 2 \exp(-(x/K_1)^{\frac{1}{\theta}})$ for all $x \geq 0$.*

(ii) *There exists a $K_2 > 0$ such that $\mathbf{E}[|X|^q]^{1/q} \leq K_2^{1/q} q^\theta$ for all $q \in (1, \infty)$.*

(iii) *There exists a $K_3 > 0$ such that $\mathbf{E}[\exp((\lambda|X|)^{\frac{1}{\theta}})] \leq \exp((\lambda K_3)^{\frac{1}{\theta}})$.*

(iv) *There exists a $K_4 > 0$ such that $\mathbf{E}[\exp((|X|/K_4)^{\frac{1}{\theta}})] \leq 2$.*

*Additionally, the constants $K_1, \dots, K_4$ differ at most by a factor that depends on $\theta$ (in particular, $K_1 = K_4$).*

**Lemma A.2.** *If $X \sim \mathsf{subW}(\theta, \sigma)$, then*

(i) $\mathbf{P}\left(|X| \leq \sigma \log^\theta(2/\delta)\right) \geq 1 - \delta,$

(ii) $\mathbf{E}[X^2] \leq 2\Gamma(2\theta + 1)\sigma^2.$

## A.2. Inequalities and lemmas

**Lemma A.3** (Smoothness inequality)**.** *If Assumption 3.1 holds, then*

$$f(x) \leq f(y) + \langle x - y, \nabla f(y) \rangle + \frac{L}{2}\|x - y\|^2$$

*for all $x, y \in \mathbf{R}^d$.*

**Lemma A.4** (Young's inequality)**.** *For any pair of vectors $x, y \in \mathbf{R}^d$,*

$$\langle x, y \rangle \leq \frac{1}{2}\|x\|^2 + \frac{1}{2}\|y\|^2.$$

**Lemma A.5** (Freedman's inequality)**.** *Consider a martingale difference sequence $\{Z_t\}_t$ adapted to a filtration $\mathcal{G}_t$ and suppose that $|Z_t| \leq \ell$ almost surely. Then with probability at least $1 - \delta$ it holds for any $\rho \in (0, 1)$ that*

$$\sum_{t=0}^{T-1} Z_t \leq \frac{\rho}{\ell} \sum_{t=0}^{T-1} \mathbf{E}[Z_t^2 \mid \mathcal{G}_t] + \frac{\ell}{\rho} \log(1/\delta).$$

# B. Proof of Theoretical Results

For notational brevity, we write $g_t^i = g_t^i(x_t)$, and denote the expectation and probability conditioned upon the $\sigma$-algebra $\mathcal{F}_t$ generated by $\{g_s\}_{s=0}^{t-1}$ by $\mathbf{E}_t[\cdot]$ and $\mathbf{P}_t\{\cdot\}$, respectively. Note that the sequences $\{x_t\}_t$ and $\{\tilde{x}_t\}_t$ are adapted to $\{\mathcal{F}_t\}_t$, and that $f$ and $\nabla f$ are measurable due to everywhere differentiability of $f$, so $\mathbf{E}_t[f(\tilde{x}_t)] = f(\tilde{x}_t)$ and $\mathbf{E}_t[\nabla f(x_t)] = \nabla f(x_t)$ almost surely. Define the virtual sequence $\{\tilde{x}_t\}_{t=0}^T$ by

$$\tilde{x}_0 = x_0, \quad \tilde{x}_1 = x_0 - \eta \sum_{i \in C_0} g_0^i, \quad \tilde{x}_{t+1} = \tilde{x}_t - \eta g_t^{i_t},$$

We first present a key inequality for our analysis, which is essentially a corollary of (Lemma 1, Mishchenko et al., 2022).

**Lemma B.1** (Virtual iterate bound)**.** *The sequences $\{x_t\}_t$ and $\{\tilde{x}_t\}_t$ satisfy*

$$\|\tilde{x}_t - x_t\| \leq \eta c \tau_C$$

*for all $t = 0, \dots, T - 1$.*

*Proof.* We have that $\tilde{x}_t = x_0 - \eta \sum_{s=1}^{t-1} g_s - \eta \sum_{i=1}^{n} g_0^i$ and $x_t = x_0 - \eta \sum_{s=1}^{t-1} g_{s-\tau(s)}^{i(s)}$. But all clipped gradients apart from the $\tau_C$ clipped gradients being computed at iteration $t$ are included in the sum $\sum_{s=1}^{t-1} g_{s-\tau(s)}^{i(s)}$, hence

$$\|\tilde{x}_t - x_t\| = \eta \left\| \sum_{s=1}^{t-1} g_s^{i(s)} - \sum_{s=1}^{t-1} g_{s-\tau(s)}^{i(s)} + \sum_{i \in C_0} g_0^i \right\| \leq \eta c \tau_C,$$

noting that $\|g_s\| \leq c$ for all $s$. $\square$

Central to our analysis is bounding the squared norm error between the clipped stochastic gradient and the full gradient, when the full gradient is small.

**Lemma B.2** (Bias and moment bounds). *Suppose Assumptions 3.1, 3.2 and 5.1 hold. Then*

$$\mathbf{E}_t \|\nabla F_{i_t}(x_t, \xi_t^{i_t}) - \nabla f(x_t)\|^2 \leq 4\sigma^2 \Gamma(2\theta + 1) + 2\zeta^2. \tag{3}$$

*Additionally, if $\|\nabla f(x_t)\| < c/2$ and $c > 2\zeta$, then*

$$\|\mathbf{E}_t g_t^{i_t} - \mathbf{clip}_c(\nabla f(x_t))\|^2 \leq \left(8\sigma^2 \Gamma(2\theta + 1) + 4\zeta^2\right) \exp\left(-\left(\frac{c - 2\zeta}{4\sigma}\right)^{\frac{1}{\theta}}\right), \tag{4}$$

*and*

$$\mathbf{E}_t \|g_t^{i_t}\|^2 \leq 24\sigma^2 \Gamma(2\theta + 1) + 8\zeta^2 + 2\|\nabla f(x_t)\|^2. \tag{5}$$

*Proof.* For (3), we first apply Young's inequality

$$\mathbf{E}_t \|\nabla F_{i_t}(x_t, \xi_t^{i_t}) - \nabla f(x_t)\|^2 \leq 2\mathbf{E}_t \|\nabla F_{i_t}(x_t, \xi_t^{i_t}) - \nabla f_{i_t}(x_t)\|^2 + 2\mathbf{E}_t \|\nabla f_{i_t}(x_t) - \nabla f(x_t)\|^2.$$

Then using (ii) from Lemma A.2 on the first term, and Assumption 5.1 on the second, we get the stated bound (3).

**Bounding $\mathbf{P}_t\{\|\nabla F_{i_t}(x_t, \xi_t^{i_t})\| \geq c\}$.** Define the indicator function $\chi_t = \mathbf{1}\{\|\nabla F_{i_t}(x_t, \xi_t^{i_t})\| \geq c\}$. We have that

$$\begin{aligned}
\mathbf{P}_t\{\chi_t = 1\} &\leq \mathbf{P}_t \left\{\|\nabla F_{i_t}(x_t, \xi_t^{i_t}) - \nabla f_{i_t}(x_t)\| + \|\nabla f_{i_t}(x_t)\| > c\right\} \\
&\leq \mathbf{P}_t \left\{\|\nabla F_{i_t}(x_t, \xi_t^{i_t}) - \nabla f_{i_t}(x_t)\| + \|\nabla f_{i_t}(x_t) - \nabla f(x_t)\| + \|\nabla f(x_t)\| > c\right\} \\
&\leq \mathbf{P}_t \left\{\|\nabla F_{i_t}(x_t, \xi_t^{i_t}) - \nabla f_{i_t}(x_t)\| > \frac{c}{2} - \zeta\right\}.
\end{aligned}$$

Applying the sub-Weibull concentration inequality (i) in Theorem A.1, we get the bound

$$\mathbf{P}_t\{\chi_t = 1\} \leq 2\exp\left(-\left(\frac{c - 2\zeta}{4\sigma}\right)^{\frac{1}{\theta}}\right). \tag{6}$$

This yields the exponential term in (4). For the polynomial bound, we bound the probability as

$$\begin{aligned}
\mathbf{P}_t\{\chi_t = 1\} &\leq \mathbf{P}_t \left\{\|\nabla F_{i_t}(x_t, \xi_t^{i_t}) - \nabla f(x_t)\| + \|\nabla f(x_t)\| > c\right\} \\
&\leq \mathbf{P}_t \left\{\|\nabla F_{i_t}(x_t, \xi_t^{i_t}) - \nabla f(x_t)\| > \frac{c}{2}\right\}.
\end{aligned}$$

and apply Markov's inequality to get

$$\mathbf{P}_t\{\chi_t = 1\} \leq \frac{4\mathbf{E}_t \|\nabla F_{i_t}(x_t, \xi_t^{i_t}) - \nabla f(x_t)\|^2}{c^2} \leq \frac{16\Gamma(2\theta + 1)\sigma^2 + 4\zeta^2}{c^2}. \tag{7}$$

**Bias bound.** Turning to the quantity $\|\mathbf{E}_t g_t^{i_t} - \mathbf{clip}_c(\nabla f(x_t))\|^2$, we first note that $\mathbf{clip}_c(\nabla f(x_t)) = \nabla f(x_t) = \mathbf{E}_t[\nabla F_{i_t}(x_t, \xi_t^{i_t})]$ since we have assumed $\|\nabla f(x_t)\| < c/2$. Thus, we have that

$$
\begin{aligned}
\|\mathbf{E}_t g_t^{i_t} - \nabla f(x_t)\|^2 &= \left\| \mathbf{E}_t \left[ \left( 1 - \frac{c}{\|\nabla F_{i_t}(x_t, \xi_t^{i_t})\|} \nabla F_{i_t}(x_t, \xi_t^{i_t}) \right) \chi_t \right] \right\|^2 \\
&\le \mathbf{E}_t \left[ \left\| \left( 1 - \frac{c}{\|\nabla F_{i_t}(x_t, \xi_t^{i_t})\|} \right) \nabla F_{i_t}(x_t, \xi_t^{i_t}) \right\| \chi_t \right]^2 \\
&= \mathbf{E}_t \left[ \left| c - \|\nabla F_{i_t}(x_t, \xi_t^{i_t})\| \right| \chi_t \right]^2 \\
&\le \mathbf{E}_t \left[ \left| \|\nabla f(x_t)\| - \|\nabla F_{i_t}(x_t, \xi_t^{i_t})\| \right| \chi_t \right]^2 .
\end{aligned}
$$

Now, using the reverse triangle inequality followed by the Cauchy-Schwarz inequality,

$$
\begin{aligned}
\mathbf{E}_t \left[ \left| \|\nabla f(x_t)\| - \|\nabla F_{i_t}(x_t, \xi_t^{i_t})\| \right| \chi_t \right]^2 &\le \mathbf{E}_t \left[ \|\nabla f(x_t) - \nabla F_{i_t}(x_t, \xi_t^{i_t})\| \chi_t \right]^2 \\
&\le \mathbf{E}_t [\|\nabla f(x_t) - \nabla F_{i_t}(x_t, \xi_t^i)\|^2] \mathbf{E}_t[\chi_t^2] \\
&\le (4\sigma^2 \Gamma(2\theta + 1) + 2\zeta^2) \mathbf{E}_t[\chi_t^2] \\
&= (4\sigma^2 \Gamma(2\theta + 1) + 2\zeta^2) \mathbf{P}_t\{\chi_t = 1\},
\end{aligned}
$$

where we applied (3) and in the last inequality we used (ii) in Lemma A.2. We arrive at the stated result (4) by applying (6).

**Second moment bound.** Turning to $\mathbf{E}_t \|g_t^{i_t}\|^2$, we have that

$$
\begin{aligned}
\mathbf{E}_t \|g_t^{i_t}\|^2 &= \mathbf{E}_t [\chi_t \|g_t^{i_t}\|^2 + (1 - \chi_t)\|g_t^{i_t}\|^2] \\
&= \mathbf{P}_t(\chi_t = 1)\mathbf{E}_t[\|g_t^{i_t}\|^2 \mid \chi_t = 1] + \mathbf{E}_t[(1 - \chi_t)\|g_t^{i_t}\|^2] \\
&= \mathbf{P}_t(\chi_t = 1)c^2 + \mathbf{E}_t \left[ (1 - \chi_t)\|\nabla F_{i_t}(x_t, \xi_t^{i_t})\|^2 \right] \\
&\le \mathbf{P}_t(\chi_t = 1)c^2 + \mathbf{E}_t \left[ \|\nabla F_{i_t}(x_t, \xi_t^{i_t})\|^2 \right] .
\end{aligned}
$$

Subsequently, we bound the term $\mathbf{E}_t \left[ \|\nabla F_{i_t}(x_t, \xi_t^{i_t})\|^2 \right]$ by applying Young's inequality and (ii) in Lemma A.2:

$$
\begin{aligned}
\mathbf{E}_t \left[ \|\nabla F_{i_t}(x_t, \xi_t^{i_t})\|^2 \right] &= \mathbf{E}_t \left[ \|(\nabla F_{i_t}(x_t, \xi_t^{i_t}) - \nabla f(x_t)) + \nabla f(x_t)\|^2 \right] \\
&\le 2\mathbf{E}_t \|\nabla F_{i_t}(x_t, \xi_t^{i_t}) - \nabla f(x_t)\|^2 + 2\mathbf{E}_t \|\nabla f(x_t)\|^2 \\
&\le 8\Gamma(2\theta + 1)\sigma^2 + 4\zeta^2 + 2\|\nabla f(x_t)\|^2 .
\end{aligned}
$$

Thus we arrive at (5) by applying (7). $\qquad \square$

## B.1. Convergence in expectation

We begin with the main descent lemma we will use in the analysis of convergence in expectation.

**Lemma B.3.** *If $f$ is $L$-smooth and $\alpha_t = \min\left\{1, \frac{c}{\|\nabla f(x_t)\|}\right\}$, then*

$$
\mathbf{E}_t f(\tilde{x}_{t+1}) - f(\tilde{x}_t) \le -\frac{\eta \alpha_t}{2} \|\nabla f(x_t)\|^2 + \frac{\eta}{2\alpha_t} \|\mathbf{E}_t g_t^{i_t} - \mathbf{clip}_c(\nabla f(x_t))\|^2 + \frac{\eta^2 L}{2}(\eta c^2 \tau_C^2 L + \mathbf{E}_t \|g_t^{i_t}\|^2). \tag{8}
$$

*Proof.* Since $f$ is $L$-smooth we have that

$$
\begin{aligned}
f(\tilde{x}_{t+1}) &\le f(\tilde{x}_t) + \langle \tilde{x}_{t+1} - \tilde{x}_t, \nabla f(\tilde{x}_t) \rangle + \frac{L}{2}\|\tilde{x}_{t+1} - \tilde{x}_t\|^2 \\
&= f(\tilde{x}_t) - \eta \langle g_t^{i_t}, \nabla f(\tilde{x}_t) \rangle + \frac{\eta^2 L}{2}\|g_t^{i_t}\|^2 .
\end{aligned} \tag{9}
$$

due to Lemma A.3. Taking the conditional expectation we have that

$$\mathbf{E}_t f(\tilde{x}_{t+1}) - f(\tilde{x}_t) \le -\eta \langle \mathbf{E}_t g_t^{i_t}, \nabla f(\tilde{x}_t) \rangle + \frac{\eta^2 L}{2} \mathbf{E}_t \|g_t^{i_t}\|^2.$$

Turning our attention to the inner product in the right-hand side, we apply Young's inequality and $L$-smoothness,

$$
\begin{aligned}
-\eta \langle \nabla f(\tilde{x}_t), \mathbf{E}_t g_t \rangle &= -\eta \langle \nabla f(x_t), \mathbf{E}_t g_t^{i_t} \rangle + \eta \langle \nabla f(x_t) - \nabla f(\tilde{x}_t), \mathbf{E}_t g_t^{i_t} \rangle \\
&\le -\eta \langle \nabla f(x_t), \mathbf{E}_t g_t^{i_t} \rangle + \frac{\eta}{2} \|\nabla f(x_t) - \nabla f(\tilde{x}_t)\|^2 + \frac{\eta}{2} \|\mathbf{E}_t g_t^{i_t}\|^2 \\
&\le -\eta \langle \nabla f(x_t), \mathbf{E}_t g_t^{i_t} \rangle + \frac{\eta L^2}{2} \|x_t - \tilde{x}_t\|^2 + \frac{\eta}{2} \|\mathbf{E}_t g_t^{i_t}\| \\
&= -\frac{\eta}{\alpha_t} \langle \mathbf{clip}_c(\nabla f(x_t)), \mathbf{E}_t g_t^{i_t} \rangle + \frac{\eta L^2}{2} \|x_t - \tilde{x}_t\|^2 + \frac{\eta}{2} \|\mathbf{E}_t g_t^{i_t}\| \\
&= -\frac{\eta \alpha_t}{2} \|\nabla f(x_t)\|^2 - \frac{\eta}{2\alpha_t} \|\mathbf{E}_t g_t^{i_t}\|^2 + \frac{\eta}{2\alpha_t} \|\mathbf{E}_t g_t^{i_t} - \mathbf{clip}_c(\nabla f(x_t))\|^2 + \frac{\eta L^2}{2} \|x_t - \tilde{x}_t\|^2 + \frac{\eta}{2} \|\mathbf{E}_t g_t^{i_t}\|^2 \\
&\le -\frac{\eta \alpha_t}{2} \|\nabla f(x_t)\|^2 + \frac{\eta}{2\alpha_t} \|\mathbf{E}_t g_t^{i_t} - \mathbf{clip}_c(\nabla f(x_t))\|^2 + \frac{\eta L^2}{2} \|x_t - \tilde{x}_t\|^2,
\end{aligned}
$$

where the last inequality is due to $\alpha_t \le 1$, so the two terms involving $\|\mathbf{E}_t g_t^{i_t}\|^2$ can be dropped. Applying Lemma B.1 we obtain the stated inequality. $\square$

**Theorem B.4.** *Suppose Assumptions 3.1 and 3.2 hold. Then there exists a constant step size $\eta$ and clipping radius $c$ such that for $\varepsilon \in (0, 1)$, we have $\frac{1}{T} \sum_{t=0}^{T-1} \mathbf{E}\|\nabla f(x_t)\| \le \varepsilon$ within*

$$\tilde{O}\left( \frac{(\sigma^2 + \zeta^2)L\Delta}{\varepsilon^4} + \frac{(\sigma + \zeta)\tau_C L\Delta}{\varepsilon^3} + \frac{\tau_C L\Delta}{\varepsilon^2} \right) \tag{10}$$

*iterations of Algorithm 1.*

*Proof.* We analyze the convergence in two cases seperately.

**Case I.** We first turn to case when $\|\nabla f(x_t)\| \le c/2$. Applying (4) and (5) from Lemma B.2 to our main descent inequality, we have

$$\mathbf{E}_t f(\tilde{x}_{t+1}) - f(\tilde{x}_t)$$

$$
\begin{aligned}
&\le -\frac{\eta}{2} \|\nabla f(x_t)\|^2 + \eta(4\sigma^2 \Gamma(2\theta + 1) + 2\zeta^2) \exp\left( -\left( \frac{c - 2\zeta}{4\sigma} \right)^{\frac{1}{\theta}} \right) + \frac{\eta^2 L}{2} \left( \eta c^2 \tau_C^2 L + 24\sigma^2 \Gamma(2\theta + 1) + 8\zeta^2 + 2\|\nabla f(x_t)\|^2 \right) \\
&= -\frac{\eta}{2} (1 - 2\eta L) \|\nabla f(x_t)\|^2 + \eta(4\sigma^2 \Gamma(2\theta + 1) + 2\zeta^2) \exp\left( -\left( \frac{c - 2\zeta}{4\sigma} \right)^{\frac{1}{\theta}} \right) + \eta^2(12\sigma^2 \Gamma(2\theta + 1) + 4\zeta^2)L + \frac{\eta^3 c^2 \tau_C^2 L^2}{2} \\
&\le -\frac{\eta}{4} \|\nabla f(x_t)\|^2 + \eta(4\sigma^2 \Gamma(2\theta + 1) + 2\zeta^2) \exp\left( -\left( \frac{c - 2\zeta}{4\sigma} \right)^{\frac{1}{\theta}} \right) + \eta^2(12\sigma^2 \Gamma(2\theta + 1) + 4\zeta^2)L + \frac{\eta^3 c^2 \tau_C^2 L^2}{2}.
\end{aligned}
$$

where in the last inequality we used $\eta \le 1/(4L)$. Taking the unconditional expectation, we have that

$$\frac{1}{4} \mathbf{E}\|\nabla f(x_t)\|^2 \le \frac{\mathbf{E}[f(\tilde{x}_t) - f(\tilde{x}_{t+1})]}{\eta} + (4\sigma^2 \Gamma(2\theta+1) + 2\zeta^2) \exp\left( -\left( \frac{c - 2\zeta}{4\sigma} \right)^{\frac{1}{\theta}} \right) + \eta(12\sigma^2 \Gamma(2\theta+1) + 4\zeta^2)L + \frac{\eta^2 c^2 \tau_C^2 L^2}{2}.$$

Summing over $t \in \mathcal{T} = \{t : \|\nabla f(x_t)\| \le c/2\}$, we obtain

$$\frac{1}{4T} \sum_{t \in \mathcal{T}} \mathbf{E}\|\nabla f(x_t)\|^2 \le$$

$$\frac{1}{T} \sum_{t \in \mathcal{T}} \left( \frac{\mathbf{E}[f(\tilde{x}_t) - f(\tilde{x}_{t+1})]}{\eta} + (4\sigma^2 \Gamma(2\theta + 1) + 2\zeta^2) \exp\left( -\left( \frac{c - 2\zeta}{4\sigma} \right)^{\frac{1}{\theta}} \right) + \eta(12\sigma^2 \Gamma(2\theta + 1) + 4\zeta^2)L + \frac{\eta^2 c^2 \tau_C^2 L^2}{2} \right).$$

**Case II.** We now turn to the case when $\|\nabla f(x_t)\| > c/2$. Using Jensen's inequality and nonexpansiveness of the clipping operator, we have

$$\|\mathbf{E}_t g_t^{i_t} - \mathbf{clip}_c(\nabla f(x_t))\|^2 \leq \mathbf{E}_t \|\nabla F_{i_t}(x_t, \xi_t^{i_t}) - \nabla f(x_t)\|^2 \leq 4\sigma^2 \Gamma(2\theta + 1) + 2\zeta^2.$$

Thus, our main inequality (8) in this case implies that

$$\mathbf{E}[f(\tilde{x}_{t+1}) - f(\tilde{x}_t)] \leq -\frac{\eta \alpha_t}{2} \mathbf{E}\|\nabla f(x_t)\|^2 + \frac{\eta}{2\alpha_t}(4\sigma^2\Gamma(2\theta+1) + 2\zeta^2) + \frac{\eta^2 L}{2}(\eta c^2 \tau_C^2 + \mathbf{E}\|g_t\|^2)$$

$$\leq -\frac{\eta \alpha_t}{2} \mathbf{E}\|\nabla f(x_t)\|^2 + \frac{\eta}{c}(4\sigma^2\Gamma(2\theta+1) + 2\zeta^2)\|\nabla f(x_t)\| + \frac{\eta^2 c^2 L}{2}(1 + \eta\tau_C^2 L)$$

where we used that $\|g_t\| \leq c$ always. Whenever $\|\nabla f(x_t)\| > c$, we have $\alpha_t = c/\|\nabla f(x_t)\|$, so if $c \geq 2\sqrt{8\Gamma(2\theta+1)\sigma^2 + 4\zeta^2}$ the inequality takes the form

$$\mathbf{E}[f(\tilde{x}_{t+1}) - f(\tilde{x}_t)] \leq -\frac{\eta c}{2}\left(1 - \frac{8\sigma^2\Gamma(2\theta+1) + 4\zeta^2}{c^2}\right)\mathbf{E}\|\nabla f(x_t)\| + \frac{\eta^2 c^2 L}{2}(1 + \eta\tau_C^2 L)$$

$$\leq -\frac{3\eta c}{8}\mathbf{E}\|\nabla f(x_t)\| + \frac{\eta^2 c^2 L}{2}(1 + \eta\tau_C^2 L)$$

Using $\eta \leq 1/(4\tau_C L)$, we arrive at

$$\frac{3c}{8}\mathbf{E}\|\nabla f(x_t)\| \leq \frac{\mathbf{E}[f(\tilde{x}_{t+1}) - f(\tilde{x}_t)]}{\eta} + \frac{\eta c^2 L}{2}(1 + \eta\tau_C^2 L).$$

Now consider the case when $c/2 < \|\nabla f(x_t)\| < c$, so $\alpha_t = 1$. In this case, we have that $-\|\nabla f(x_t)\| \leq -c/2$ and $1 \leq \|\nabla f(x_t)\|/c$. Thus, similarly as in the previous case, our main inequality (8) implies that

$$\mathbf{E}_t f(\tilde{x}_{t+1}) - f(\tilde{x}_t) \leq -\frac{\eta c}{4}\|\nabla f(x_t)\| + \frac{\eta}{c}2\Gamma(2\theta+1)\sigma^2\|\nabla f(x_t)\| + \frac{\eta^2 c^2 L}{2}(1 + \eta\tau_C^2 L)$$

$$\leq -\frac{\eta c}{4}\left(1 - \frac{8\sigma^2\Gamma(2\theta+1)}{c^2}\right)\|\nabla f(x_t)\| + \frac{\eta^2 c^2 L}{2}(1 + \eta\tau_C^2 L)$$

$$\leq -\frac{\eta c}{8}\|\nabla f(x_t)\| + \frac{\eta^2 c^2 L}{2}(1 + \eta\tau_C^2 L).$$

Taking the unconditional expectation, we arrive at the inequality

$$\frac{c}{8}\mathbf{E}\|\nabla f(x_t)\| \leq \frac{\mathbf{E}[f(\tilde{x}_t) - f(\tilde{x}_{t+1})]}{\eta} + \frac{\eta c^2 L}{2}(1 + \eta\tau_C^2 L).$$

Thus we have that

$$\frac{1}{8T}\sum_{t\in\mathcal{T}^c} c\mathbf{E}\|\nabla f(x_t)\| = \frac{1}{T}\sum_{t\in\mathcal{T}^c}\left(\frac{\mathbf{E}[f(\tilde{x}_t) - f(\tilde{x}_{t+1})]}{\eta} + \frac{\eta c^2 L}{2}(1 + \eta\tau_C^2 L)\right)$$

**Choosing the parameters.** Putting together the two cases, we have that

$$\frac{1}{8T}\left(\sum_{t\in\mathcal{T}}\mathbf{E}\|\nabla f(x_t)\|^2 + \sum_{t\in\mathcal{T}^c}c\mathbf{E}\|\nabla f(x_t)\|\right) \leq \frac{1}{T}\sum_{t=0}^{T-1}\frac{\mathbf{E}[f(\tilde{x}_t) - f(\tilde{x}_{t+1})]}{\eta} + (4\sigma^2\Gamma(2\theta+1) + 2\zeta^2)\exp\left(-\left(\frac{c-2\zeta}{4\sigma}\right)^{\frac{1}{\theta}}\right)$$

$$+ \eta(12\sigma^2\Gamma(2\theta+1) + 4\zeta^2)L + \frac{\eta^2 c^2\tau_C^2 L^2}{2} + \frac{\eta c^2 L}{2}$$

$$= O\left(\frac{\Delta}{\eta T} + (\sigma^2 + \zeta^2)\left(\eta L + \exp\left(-\left(\frac{c-2\zeta}{4\sigma}\right)^{\frac{1}{\theta}}\right)\right) + \eta^2 c^2\tau_C^2 L^2 + \eta c^2 L\right).$$

This means that both

$$\frac{1}{T}\sum_{t\in\mathcal{T}}\mathbf{E}\|\nabla f(x_t)\| = O\left(\sqrt{\frac{\Delta}{\eta T}} + \sqrt{\eta(\sigma^2 + \zeta^2)L} + \sqrt{\sigma^2 + \zeta^2}\exp\left(-\left(\frac{c-2\zeta}{4\sigma}\right)^{\frac{1}{\theta}}\right) + \eta c\tau_C L + \sqrt{\eta c^2 L}\right)$$

and

$$\frac{1}{T} \sum_{t \in \mathcal{T}^c} \mathbf{E}\|\nabla f(x_t)\| = O\left(\frac{\Delta}{\eta cT} + \frac{\sigma^2 + \zeta^2}{c}\left(\eta L + \exp\left(-\left(\frac{c - 2\zeta}{4\sigma}\right)^{\frac{1}{\theta}}\right)\right) + \eta^2 c\tau_C^2 L^2 + \eta cL\right).$$

We choose the clipping radius to be $c = 2\sqrt{8\Gamma(2\theta + 1)\sigma^2}\log^\theta T + 4\zeta$. Assuming $T$ is large enough so that $c \geq 1$, we have that

$$\frac{1}{T} \sum_{t=0}^{T-1} \mathbf{E}\|\nabla f(x_t)\| = \widetilde{O}\left(\sqrt{\frac{\Delta}{\eta T}} + \sqrt{\eta(\sigma^2 + \zeta^2)L} + \eta(\sigma + \zeta)\tau_C L \log^\theta T + \sqrt{\frac{\sigma^2 + \zeta^2}{T}}\right).$$

Then choosing the step size to be

$$\eta = \min\left\{\frac{1}{4\tau_C L}, \left(\frac{\Delta}{(\sigma^2 + \zeta^2)LT}\right)^{1/2}, \left(\frac{\Delta}{(\sigma^2 + \zeta^2)\tau_C^2 L^2 T}\right)^{1/3}\right\},$$

we obtain

$$\frac{1}{T} \sum_{t=0}^{T-1} \mathbf{E}\|\nabla f(x_t)\| = \widetilde{O}\left(\sqrt{\frac{\tau_C L\Delta}{T}} + \left(\frac{(\sigma^2 + \zeta^2)L\Delta}{T}\right)^{1/4} + \left(\frac{(\sigma + \zeta)\tau_C L\Delta}{T}\right)^{1/3} + \sqrt{\frac{\sigma^2 + \zeta^2}{T}}\right).$$

Hence, we require

$$\widetilde{O}\left(\frac{(\sigma^2 + \zeta^2)L\Delta}{\varepsilon^4} + \frac{(\sigma + \zeta)\tau_C L\Delta}{\varepsilon^3} + \frac{\tau_C L\Delta}{\varepsilon^2}\right).$$

iteration to reach $\varepsilon$-stationarity. $\qquad\square$

## B.2. Convergence with high probability

**Lemma B.5.** *If $f$ is $L$-smooth and $\alpha_t = \min\{1, c/\|\nabla f(x_t)\|\}$, then*

$$\begin{aligned}
f(\tilde{x}_{t+1}) - f(\tilde{x}_t) &\leq -\frac{\eta}{2}\|\nabla f(x_t)\|^2 + \frac{\eta}{2}\|\mathbf{E}_t g_t^{i_t} - \nabla f(x_t)\|^2 + \frac{\eta^3 c^2\tau_C^2 L^2}{2} + \frac{\eta^2 L}{2}\|g_t^{i_t}\|^2 \\
&\quad - \eta\langle\nabla f(x_t), g_t^{i_t} - \mathbf{E}_t g_t^{i_t}\rangle - \eta\langle\nabla f(\tilde{x}_t) - \nabla f(x_t), g_t^{i_t} - \mathbf{E}_t g_t^{i_t}\rangle.
\end{aligned} \tag{11}$$

*Proof.* Due to $L$-smoothness we have that

$$\begin{aligned}
f(\tilde{x}_{t+1}) &\leq f(\tilde{x}_t) + \langle\tilde{x}_{t+1} - \tilde{x}_t, \nabla f(\tilde{x}_t)\rangle + \frac{L}{2}\|\tilde{x}_{t+1} - \tilde{x}_t\|^2 \\
&= f(\tilde{x}_t) - \eta\langle g_t^{i_t}, \nabla f(\tilde{x}_t)\rangle + \frac{\eta^2 L}{2}\|g_t^{i_t}\|^2.
\end{aligned}$$

due to Lemma A.3. We first expand the inner product according to

$$\begin{aligned}
-\eta\langle\nabla f(\tilde{x}_t), g_t^{i_t}\rangle = &-\eta\langle\nabla f(x_t), \mathbf{E}_t g_t^{i_t}\rangle && \text{(deterministic descent)} \\
&- \eta\langle\nabla f(\tilde{x}_t) - \nabla f(x_t), \mathbf{E}_t g_t^{i_t}\rangle && \text{(delay error)} \\
&\underbrace{-\eta\langle\nabla f(x_t), g_t^{i_t} - \mathbf{E}_t g_t^{i_t}\rangle - \eta\langle\nabla f(\tilde{x}_t) - \nabla f(x_t), g_t^{i_t} - \mathbf{E}_t g_t^{i_t}\rangle}_{Z_t}, && \text{(martingale difference)}
\end{aligned}$$

where we note that the last two terms form a martingale difference sequence $Z_t$ adapted to $\mathcal{G}_t = \mathcal{F}_{t+1}$. The first inner product in the right-hand side can be written as

$$-\eta\langle\nabla f(x_t), \mathbf{E}_t g_t^{i_t}\rangle = -\frac{\eta}{2}\|\nabla f(x_t)\|^2 - \frac{\eta}{2}\|\mathbf{E}_t g_t^{i_t}\|^2 + \frac{\eta}{2}\|\mathbf{E}_t g_t^{i_t} - \nabla f(x_t)\|^2. \tag{12}$$

Using Young's inequality (Lemma A.4) on the second inner product gives the bound

$$-\eta\langle\nabla f(\tilde{x}_t) - \nabla f(x_t), \mathbf{E}_t g_t^{i_t}\rangle \leq \frac{\eta}{2}\|\nabla f(\tilde{x}_t) - \nabla f(x_t)\|^2 + \frac{\eta}{2}\|\mathbf{E}_t g_t^{i_t}\|^2. \tag{13}$$

Moreover, due to $L$-smoothness of $f$ (Assumption 3.1) and Lemma B.1, we have

$$\frac{\eta}{2}\|\nabla f(\tilde{x}_t) - \nabla f(x_t)\|^2 \le \frac{\eta L^2}{2}\|\tilde{x}_t - x_t\|^2 \le \frac{\eta^3 c^2 L^2}{2}$$

By subtracting $f(\tilde{x}_t)$ from both sides of (9), we arrive at the stated inequality by noting that the $\frac{\eta}{2}\mathbf{E}_t\|g_t\|^2$ terms in (12) and (13) cancel. $\qquad\square$

**Theorem B.6.** *Suppose Assumptions 3.1 and 3.2 hold. Then there exists a constant step size $\eta$ and clipping radius $c$ such that for $\varepsilon \in (0, 1)$ and failure probability $\delta \in (0, 1)$, we have $\mathbf{P}\left(\frac{1}{T}\sum_{t=0}^{T-1}\|\nabla f(x_t)\| \le \varepsilon\right) \ge 1 - \delta$ within*

$$\widetilde{O}\left(\frac{\sigma^2 \log^{2\theta}(1/\delta)}{\varepsilon^4} + \frac{\sigma\tau_C \log^{\theta}(1/\delta)}{\varepsilon^3} + \frac{\tau_C}{\varepsilon^2}\right) \tag{14}$$

*iterations of Algorithm 1.*

*Proof.* Similarly as before, we analyze the convergence in two different cases seperately.

**Case I.** We begin by treating the case when $\|\nabla f(x_t)\| \le c/2$, in which we will show that the martingale difference sequence concentrates. First, we denote the martingale difference term

$$Z_t = -\eta\langle\nabla f(x_t), g_t^{i_t} - \mathbf{E}_t g_t^{i_t}\rangle - \eta\langle\nabla f(\tilde{x}_t) - \nabla f(x_t), g_t^{i_t} - \mathbf{E}_t g_t^{i_t}\rangle$$

and the indicator function $\psi_t = \{\|\nabla f(x_t)\| \le c/2\}$ Since clipping ensures $\|g_t^{i_t}\| \le c$ almost surely, we can bound $|Z_t|$ uniformly for $t \in \mathcal{T}$:

$$\begin{aligned}
|\psi_t Z_t| &\le \eta(\|\nabla f(x_t)\| + \|\nabla f(\tilde{x}_t) - \nabla f(x_t)\|)\left(\|g_t^{i_t}\| + \|\mathbf{E}_t g_t^{i_t}\|\right) \\
&\le 2\eta\left(\frac{c}{2} + \eta c\tau_C L\right)c \\
&\le \frac{3\eta c^2}{4}
\end{aligned}$$

where in the second inequality we use that $\eta \le 1/(4\tau_C L)$. By applying the Cauchy-Schwarz inequality and Young's inequality we can also bound the conditional second moment of $Z_t$:

$$\begin{aligned}
\mathbf{E}[Z_t^2 \mid \mathcal{F}_t] &\le \eta^2\|\nabla f(x_t) + \nabla f(\tilde{x}_t) - \nabla f(x_t)\|^2\|g_t^{i_t} - \mathbf{E}_t g_t^{i_t}\|^2 \\
&\le \eta^2\left(2\|\nabla f(x_t)\|^2 + 2\|\nabla f(\tilde{x}_t) - \nabla f(x_t)\|^2\right)\left(2\|g_t^{i_t}\|^2 + 2\|\mathbf{E}_t g_t^{i_t}\|^2\right) \\
&\le 4\eta^2 c^2\left(\|\nabla f(x_t)\|^2 + \eta^2 c^2 \tau_C^2 L^2\right).
\end{aligned}$$

So applying Freedman's inequality (Lemma A.5) with $\rho = 3/16$, we have with probability at least $1 - \delta/2$ that

$$\begin{aligned}
\sum_{t\in\mathcal{T}} Z_t &\le \frac{4\rho\eta^2 c^2}{3\eta c^2}\sum_{t\in\mathcal{T}}\left(\|\nabla f(x_t)\|^2 + \eta^2 c^2 \tau_C^2 L^2\right) + \frac{3\eta c^2}{4\rho}\log\frac{2}{\delta} \\
&\le \frac{1}{4}\sum_{t\in\mathcal{T}}\left(\eta\|\nabla f(x_t)\|^2 + \eta^3 c^2 \tau_C^2 L^2\right) + 4\eta c^2 \log\frac{2}{\delta}.
\end{aligned}$$

Thus, summing (11) over $t \in \mathcal{T}$, we have

$$\begin{aligned}
\sum_{t\in\mathcal{T}}(f(\tilde{x}_{t+1}) - f(\tilde{x}_t)) &\le \sum_{t\in\mathcal{T}}\left(-\frac{\eta}{2}\|\nabla f(x_t)\|^2 + \frac{\eta}{2}\|\mathbf{E}_t g_t^{i_t} - \nabla f(x_t)\|^2 + \frac{\eta^3 c^2 \tau_C^2 L^2}{2} + \frac{\eta^2 L}{2}\|g_t^{i_t}\|^2 + Z_t\right) \\
&\le \sum_{t\in\mathcal{T}}\left(-\frac{\eta}{4}\|\nabla f(x_t)\|^2 + \frac{\eta}{2}\|\mathbf{E}_t g_t^{i_t} - \nabla f(x_t)\|^2 + \frac{3\eta^3 c^2 \tau_C^2 L^2}{4} + \frac{\eta^2 c^2 L}{2}\right) + 4\eta c^2 \log\frac{2}{\delta}
\end{aligned}$$

under the same probability. Applying (4) from Lemma (B.2) to the clipping error term $\|\mathbf{E}_t g_t^{i_t} - \nabla f(x_t)\|^2$ and dividing by $\eta$, we have

$$\frac{1}{4} \sum_{t \in \mathcal{T}} \|\nabla f(x_t)\|^2$$

$$\leq \sum_{t \in \mathcal{T}} \left( \frac{f(\tilde{x}_t) - f(\tilde{x}_{t+1})}{\eta} + (4\sigma^2 \Gamma(2\theta + 1) + 2\zeta^2) \exp\left( -\left( \frac{c - 2\zeta}{4\sigma} \right)^{\frac{1}{\theta}} \right) + \frac{3\eta^2 c^2 \tau_C^2 L^2}{4} + \frac{\eta c^2 L}{2} \right) + 4c^2 \log \frac{2}{\delta}$$

with probability at least $1 - \delta/2$.

**Case II.** We consider the case $\|\nabla f(x_t)\| > c/2$. By smoothness, we have

$$f(\tilde{x}_{t+1}) - f(\tilde{x}_t) \leq -\frac{\eta \alpha_t}{2} \|\nabla f(x_t)\|^2 + \frac{\eta}{2\alpha_t} \|g_t^{i_t} - \mathbf{clip}_c(\nabla f(x_t))\|^2 + \frac{\eta^3 c^2 \tau_C^2 L^2}{2} + \frac{\eta^2 c^2 L}{2}$$

$$\leq -\frac{\eta \alpha_t}{2} \|\nabla f(x_t)\|^2 + \frac{\eta}{2\alpha_t} \|\nabla F_{i_t}(x_t, \xi_t^{i_t}) - \nabla f(x_t)\|^2 + \frac{\eta^3 c^2 \tau_C^2 L^2}{2} + \frac{\eta^2 c^2 L}{2}.$$

where the second inequality holds due to nonexpansiveness of the clipping operator. Then, under sub-Weibull noise (Assumption 3.2),

$$\sum_{t \in \mathcal{T}^c} \|\nabla F_{i_t}(x_t, \xi_t^{i_t}) - \nabla f(x_t)\|^2 \leq \sum_{t \in \mathcal{T}^c} (2\|\nabla F_{i_t}(x_t, \xi_t^{i_t}) - \nabla f_{i_t}(x_t)\|^2 + 2\|\nabla f_{i_t}(x_t) - \nabla f(x_t)\|^2)$$

$$\leq 2\sigma^2 \log^{2\theta}(2T/\delta) + 2\zeta^2$$

with probability at least $1 - \delta/2$. Thus for all $t \in \mathcal{T}^c$, we have under the same probability that if $\|\nabla f(x_t)\| \leq c$

$$f(\tilde{x}_{t+1}) - f(\tilde{x}_t) \leq -\frac{\eta \alpha_t}{2} \|\nabla f(x_t)\|^2 + \frac{\eta}{\alpha_t} (\sigma^2 \log^{2\theta}(4T/\delta) + \zeta^2) + \frac{\eta^3 c^2 \tau_C^2 L^2}{2} + \frac{\eta^2 c^2 L}{2}$$

$$\leq -\frac{\eta c}{4} \|\nabla f(x_t)\| + 2\frac{\eta}{c} (\sigma^2 \log^{2\theta}(4T/\delta) + \zeta^2) \|\nabla f(x_t)\| + \frac{\eta^3 c^2 \tau_C^2 L^2}{2} + \frac{\eta^2 c^2 L}{2}$$

$$\leq -\frac{\eta c}{4} \left( 1 - \frac{8(\sigma^2 \log^{2\theta}(4T/\delta) + \zeta^2)}{c^2} \right) \|\nabla f(x_t)\| + \frac{\eta^3 c^2 \tau_C^2 L^2}{2} + \frac{\eta^2 c^2 L}{2},$$

whereas if $\|\nabla f(x_t)\| \geq c$,

$$f(\tilde{x}_{t+1}) - f(\tilde{x}_t) \leq -\frac{\eta \alpha_t}{2} \|\nabla f(x_t)\|^2 + \frac{\eta}{\alpha_t} (\sigma^2 \log^{2\theta}(2/\delta) + \zeta^2) + \frac{\eta^3 c^2 \tau_C^2 L^2}{2} + \frac{\eta^2 c^2 L}{2}$$

$$\leq -\frac{\eta c}{2} \|\nabla f(x_t)\| + \frac{\eta}{c} (\sigma^2 \log^{2\theta}(2/\delta) + \zeta^2) \|\nabla f(x_t)\| + \frac{\eta^3 c^2 \tau_C^2 L^2}{2} + \frac{\eta^2 c^2 L}{2}$$

$$\leq -\frac{\eta c}{2} \left( 1 - \frac{2(\sigma^2 \log^{2\theta}(2/\delta) + \zeta^2)}{c^2} \right) \|\nabla f(x_t)\| + \frac{\eta^3 c^2 \tau_C^2 L^2}{2} + \frac{\eta^2 c^2 L}{2}.$$

Hence, choosing $c \geq 4\sqrt{\sigma^2 \log^{2\theta}(2/\delta) + \zeta^2}$, we have

$$\frac{1}{8T} \sum_{t \in \mathcal{T}^c} c\|\nabla f(x_t)\| \leq \sum_{t \in \mathcal{T}^c} \left( \frac{f(\tilde{x}_{t+1}) - f(\tilde{x}_t)}{\eta} + \frac{\eta^2 c^2 \tau_C^2 L^2}{2} + \frac{\eta c^2 L}{2} \right)$$

with probability at least $1 - \delta/2$.

**Choosing the parameters.** Putting the two cases together, we have that

$$\frac{1}{8T} \left( \sum_{t \in \mathcal{T}} \|\nabla f(x_t)\|^2 + \sum_{t \in \mathcal{T}^c} c\|\nabla f(x_t)\| \right) \leq \frac{1}{T} \sum_{t=0}^{T-1} \frac{f(\tilde{x}_t) - f(\tilde{x}_{t+1})}{\eta} + \frac{3\eta^2 c^2 \tau_C^2 L^2}{4} + \frac{\eta c^2 L}{2}$$

$$+ (4\sigma^2 \Gamma(2\theta + 1) + 2\zeta^2) \exp\left( -\left( \frac{c - 2\zeta}{4\sigma} \right)^{\frac{1}{\theta}} \right) + 4c^2 \log \frac{2}{\delta}$$

with probability atleast $1 - \delta$. Using the same technique as in the proof of Theorem B.4, we have that

$$\frac{1}{T} \sum_{t=0}^{T-1} \|\nabla f(x_t)\| = O\left(\frac{\Delta}{\eta c T} + \sqrt{\frac{\Delta}{\eta T}} + \sqrt{\eta c^2 L} + \eta c \tau_C L + \sqrt{\sigma^2 + \zeta^2} \exp\left(-\frac{1}{2}\left(\frac{c - 2\zeta}{4\sigma}\right)^{\frac{1}{\theta}}\right) + \sqrt{\frac{c^2 \log(1/\delta)}{T}}\right)$$

with probability at least $1 - \delta$. Then if

$$c = \max\left\{4\sqrt{\sigma^2 \log^{2\theta}(2/\delta) + \zeta^2}, 4\sigma \log^\theta T + 2\zeta\right\}$$

and assuming $T$ is large enough that $c \geq 1$, we have

$$\frac{1}{T} \sum_{t=0}^{T-1} \|\nabla f(x_t)\| = \widetilde{O}\left(\sqrt{\frac{\Delta}{\eta T}} + \left(\sqrt{\eta(\sigma^2 + \zeta^2)L} + \eta(\sigma + \zeta)\tau_C L\right) \log^\theta(T/\delta) + \sqrt{\frac{\sigma^2 \log^{2\theta+1}(1/\delta) + \zeta^2}{T}}\right)$$

with probability at least $1 - \delta$. Then choosing

$$\eta = \min\left\{\frac{1}{4\tau_C L}, \left(\frac{\Delta}{(\sigma^2 + \zeta^2)\log^{2\theta}(T/\delta)LT}\right)^{1/2}, \left(\frac{\Delta}{(\sigma^2 + \zeta^2)\tau_C^2 L^2 T}\right)^{1/3}\right\},$$

we have

$$\frac{1}{T} \sum_{t=0}^{T-1} \|\nabla f(x_t)\|$$

$$= \widetilde{O}\left(\left(\frac{(\sigma^2 + \zeta^2)\log^{2\theta}(T/\delta)L\Delta}{T}\right)^{1/4} + \left(\frac{(\sigma + \zeta)\tau_C \log^\theta(T/\delta)L\Delta}{T}\right)^{1/3} + \left(\frac{\tau_C L\Delta}{T}\right)^{1/2} + \left(\frac{\sigma^2 \log^{2\theta+1}(1/\delta) + \zeta^2}{T}\right)^{1/2}\right)$$

with the same probability. Then, for $\varepsilon \in (0, 1)$, it holds that $\mathbf{P}(\frac{1}{T} \sum_{t=0}^{T-1} \|\nabla f(x_t)\| \leq \varepsilon) \geq 1 - \delta$ after

$$\widetilde{O}\left(\frac{(\sigma^2 + \zeta^2)\log^{2\theta}(1/\delta)L\Delta}{\varepsilon^4} + \frac{(\sigma + \zeta)\tau_C \log^\theta(1/\delta)L\Delta}{\varepsilon^3} + \frac{\tau_C L\Delta}{\varepsilon^2} + \frac{\sigma^2 \log^{2\theta+1}(1/\delta) + \zeta^2}{\varepsilon^2}\right)$$

iterations.

$\square$

