# OpenReview forum: "Clipping Makes Distributed and Federated Asynchronous SGD Robust to Stragglers"
_ICML.cc/2026/Conference — ICML 2026 regular_

### Official Review · Reviewer_Viw8 · 2026-03-07

**Soundness:** 2
**Presentation:** 2
**Significance:** 3
**Originality:** 3
**Overall Recommendation:** 4
**Confidence:** 4

**Summary:**

This paper analyzes Clipped Asynchronous SGD for smooth, non-convex optimization. The authors prove that the method converges for both homogeneous and heterogeneous data cases.

A key result is that their iteration complexity does not depend on the maximum delay, which is an improvement over standard asynchronous SGD analysis. Additionally, the authors provide high-probability convergence guarantees under a sub-Weibull noise assumption.

**Compliance With Llm Reviewing Policy:**

Affirmed.

**Final Justification:**

The authors acknowledged the concerns raised and expressed willingness to revise the paper accordingly. Therefore, I am increasing my score, with the expectation that these changes will be incorporated in the camera-ready version of the paper.

**Key Questions For Authors:**

See the weaknesses listed above.

**Limitations:**

See the weaknesses listed above.

**Strengths And Weaknesses:**

### Strengths

1. **First analysis of clipped ASGD**: To my knowledge, this is the first theoretical analysis of the clipped version of asynchronous SGD.

2. **High-probability results**: This appears to be the first work to provide high-probability convergence guarantees in an asynchronous setting, which is a useful addition to standard average-case analysis.


### Weaknesses

1. **Missing State-of-the-Art (SOTA) references**: The authors repeatedly claim that delay-adaptive ASGD [1-2] is the state of the art. However, recent work [3] has shown this is not the case and proposed optimal parallel methods. These results have also been extended to the asynchronous case for both homogeneous [4] and heterogeneous [5] data cases. The authors should acknowledge these modern benchmarks, compare their results against them, and ideally include them in the experiments to avoid misleading claims about what is currently the "best" method.

2. **Lack of time complexity analysis**: Evaluating asynchronous methods based only on iteration complexity is misleading. As shown in [3-5], asynchronous methods can actually have worse iteration complexity than synchronous ones; their real benefit is in time complexity.

3. **Suboptimal results**: Even if a time complexity analysis were performed, the results of this method would not be optimal compared to the current state of the art [3-5]. The authors should at least mention this gap in optimality somewhere in the paper.

---

[1] Anastasiia Koloskova, Sebastian U Stich, and Martin Jaggi. "Sharper convergence guarantees for asynchronous SGD for distributed and federated learning". Advances in Neural Information Processing Systems, 35:17202–17215, 2022.

[2] Konstantin Mishchenko, Francis Bach, Mathieu Even, and Blake E Woodworth. "Asynchronous SGD beats minibatch SGD under arbitrary delays". Advances in Neural Information Processing Systems, 35:420–433, 2022.

[3] Alexander Tyurin and Peter Richt´arik. "Optimal time complexities of parallel stochastic optimization methods under a fixed computation model". Advances in Neural Information Processing Systems, 36, 2024

[4] Artavazd Maranjyan, Alexander Tyurin, and Peter Richt´arik. "Ringmaster ASGD: The first Asynchronous SGD with optimal time complexity". International Conference on Machine Learning, 2025

[5] Artavazd Maranjyan and Peter Richt´arik. "Ringleader ASGD: The first Asynchronous SGD with optimal time complexity under data heterogeneity". arXiv preprint arXiv:2509.22860, 2025

---

> ### Author Rebuttal · Authors · 2026-03-30
>
> Thank you for the careful reading and for highlighting both the novelty of the Clipped ASGD analysis and the high-probability contribution.
> ### **Strengths**
> We would also like to highlight another contribution in providing a theoretical explanation of previously unexplained behavior in real large scale asynchronous deep learning training (see the third paragraph of Section 1).
> ### **Weaknesses**
> We agree that the paper should acknowledge the recent asynchronous literature more carefully. In particular, we will revise the related work and theorem discussion so that the paper does not suggest that Delay-adaptive ASGD is the strongest current baseline in general, and we will discuss the relation to the more recent optimal-time-complexity ASGD methods in both the homogeneous and heterogeneous settings. Moreover, we will include Ringmaster and Ringleader ASGD in the experiments. To this end, we have included new experimental results below with Ringmaster ASGD, as well as Ringleader ASGD in the response to Reviewer W8P9.
>
> We also agree that iteration complexity alone does not fully capture the value of asynchronous methods, whose practical advantage is fundamentally about wall-clock time. We have included a discussion of the time complexity of Clipped ASGD in point 2 of the response to Reviewer dY21. We will further clarify that the present theory focuses on the optimization effect of clipping, namely the removal of the dependence on the maximum delay from the oracle complexity. We will make this scope clearer in the revision and add an explicit discussion of time complexity.
>
> We also agree that our method is not claimed to be optimal in time complexity compared with the most recent ASGD literature, and we will state this explicitly. The goal of the paper is to better understand the role of clipping in asynchronous optimization, in particular its robustness to stale updates, rather than to claim a universally best asynchronous algorithm.
>
> ### **New experimental results**
> We have rerun the CIFAR10 experiments (with the same setup as in the paper), now with Ringmaster ASGD for delay thresholds $R =8,16,32$. In Tables 1 and 2 we show the results for delay factors $D =4,8$, now showing simulated wall-clock to reach 80% test accuracy rather than iterations. In order of best to worst performance in these experiments, we have Clipped ASGD (clipping radius $c=1$), Ringmaster ASGD (delay threshold $R=16$), Delay-adaptive ASGD, and Vanilla ASGD.
>
> **Table 1.** Simulated wall-clock time to reach 80% test accuracy with delay factor $D=4$. The maximum wall-clock time is 5,000 time units. The average time per oracle call was 0.108 time units.
>
> | Step size | Vanilla | Delay-adaptive | Ringmaster R = 8 | Ringmaster R = 16 | Ringmaster R = 32| Clipped c = 1.0 | Clipped c = 2.0 | Clipped c = 4.0 |
> |-|-|-|-|-|-|-|-|-|
> | 0.000977  | 5000| 5000| 5000| 5000| 5000| 5000| 5000| 5000|
> | 0.001953  | 5000| 5000| 5000| 5000| 5000| 5000| 5000| 5000|
> | 0.003906  | 3352| 4325| 5000| 4109| 4433| 5406| 4109| 4109|
> | 0.007812  | 2272| 2812| 3784| 2920| 3028| 3568| 2272| 2380|
> | 0.015625  | 1839| 2272| 2704| 1947| 2163| 1947| 1731| 1839|
> | 0.031250  | 1623| 1839| 2055| 1731| 1731| 1515| 1623| 1731|
> | 0.062500  | 1839| 1515| 1947| 1406| 1406| **1298**| 1406| 1623|
> | 0.125000  | 5000| 5000| 1623| 1623| 2055| 1406| 1839| 1731|
> | 0.250000  | 5000| 5000| 5000| 5000| 5000| 5000| 5000| 5000|
> | 0.500000  | 5000| 5000| 5000| 5000| 5000| 5000| 5000| 5000|
>
> **Table 2.** Simulated wall-clock time to reach 80% test accuracy with delay factor $D=8$. The maximum wall-clock time is 6,000 units. The average time per oracle call was 0.123 time units.
>
> | Step size | Vanilla | Delay-adaptive | Ringmaster R = 8 | Ringmaster R = 16 | Ringmaster R = 32 | Clipped c = 1.0 | Clipped c = 2.0 | Clipped c = 4.0 |
> |-|-|-|-|-|-|-|-|-|
> | 0.000977|6000| 6000| 6000| 6000| 6000| 6000| 6000| 6000|
> | 0.001953|5048| 5909| 6000| 6000| 6000| 6000| 6000| 6000|
> | 0.003906|3201| 4063| 4555| 4678| 4309| 6000| 6000| 3939|
> | 0.007812|2094| 2832| 3078| 2955| 3324| 6000| 3324| 2585|
> | 0.015625|1848| 2216| 2463| 2463| 2093| 3448| 2094| 1724|
> | 0.031250|6000| 1478| 1848| 1724| 1724| 1970| 1478| 1601|
> | 0.062500|6000| 1601| 2093| 1478| 1724| 1232| 1109| 1478|
> | 0.125000|6000| 6000| 2848| 6000| 1724| **985**| 1355| 2709|
> | 0.250000|6000| 6000| 6000| 6000| 6000| 1232| 4924| 6000|
> | 0.500000|6000| 6000| 6000| 6000| 6000| 6000| 6000| 6000|

---

> > ### Author Rebuttal · Reviewer_Viw8 · 2026-04-03
> >
> > I am glad to see that the authors acknowledge my concerns and are willing to revise the paper to address them. I appreciate their responsiveness and their commitment to improving the presentation and correctness of the work.
> >
> > Based on this, I am willing to increase my score. However, this is somewhat conditional on the expectation that these promised changes will indeed be incorporated in the camera-ready version.

---

> > > ### Author Response · Authors · 2026-04-05
> > >
> > > Thank you again for your thorough reading of our paper and the constructive feedback. We also thank you raising your score.
> > >
> > > Since the discussion phase does not allow us to upload a revised manuscript, we want to confirm clearly that we will incorporate the changes we committed to in the rebuttal. In response to the points you have raised, we will in the camera-ready version:
> > > 1. include an explicit discussion on the time-complexity of Clipped ASGD, and compare with the recent time-optimal methods Ringmaster and Ringleader ASGD [4,5], and clarify the optimality gap of Clipped ASGD
> > > 2. remove any language suggesting Delay-adaptive ASGD is the state-of-the-art,
> > > 3. include Ringmaster and Ringleader ASGD in the experiments on wall-clock time to reach target test accuracy.

---

### Official Review · Reviewer_W8P9 · 2026-03-11

**Soundness:** 3
**Presentation:** 3
**Significance:** 3
**Originality:** 3
**Overall Recommendation:** 4
**Confidence:** 4

**Summary:**

The paper provides a theoretical reasoning for the empirical observation that clipping gradients stabilizes the training in federated asynchronous stochastic gradient descent. More specifically, it investigates the effect of gradient clipping in oracle complexities and shows that the oracle complexity no longer depends on maximum delay, but only on concurrency.

To show this dependence, the paper provides both in-expectation and high-probability theoretical guarantees. In addition, the paper makes a more realistic assumption about gradient noise (a sub-Weibull distribution), motivated by previous studies and its own observations on CIFAR10 data trained with ResNet18. The paper claims to be the first to give high-probability convergence guarantees for asynchronous federated SGD.

**Compliance With Llm Reviewing Policy:**

Affirmed.

**Final Justification:**

The Authors addressed my concerns, and the paper is therefore worthy of the score assigned.
The score given is conditional on the authors making the promised changes to the camera-ready version.

**Key Questions For Authors:**

Did you run numerical experiments on heterogeneous datasets? If not, please consider doing so and reporting the model for data heterogeneity used.

Could you (or have you already) run more experiments with different NN architectures and different data distributions to validate the derived theoretical results?

Please provide more details on the experimental setup and consider uploading the code to a public repository for duplication of the results.

**Limitations:**

The reviewer does not think that the paper poses a negative societal impact. However, this is not discussed in the paper.

**Strengths And Weaknesses:**

Strengths:
The paper is well-written. It has a smooth flow of information, allowing the reader to connect ideas and appreciate the contributions.
The contributions are solid and motivated by theoretical and empirical results.

The main contribution of the paper is to provide a theoretical explanation of an empirical observation and to close the gap between theory and practice. The provided theoretical guarantees are correct, to the extent of the reviewer's knowledge.

The paper justifies its assumptions with empirical evidence (Figure 1) and by citing related papers.

Weaknesses:
The paper has minor weaknesses, as outlined below.

Experiments are restricted to the homogeneous case (except if the Shakespeare dataset used has a heterogeneous distribution).
Showing the empirical convergence in heterogeneous settings would strengthen the paper.

The notation used for Algorithm 1 can be better explained.
For instance, this can be done by providing an intuitive explanation of the algorithm before presenting it.
The current notation is also cumbersome. In particular, in equation 2, it is not whether the gradients are computed on a virtual sequence of models or on actual models. The same holds for the proofs. A better explanation is required.

The asynchronous SGD can be explained in the problem setup prior to the algorithms 1 and 2, so that the reader is made familiar with the setting and the notation used for the gradients.

In the Theorems, ..\epsilon.. is used to denote the error bound and called the accuracy. This naming convention leads to ambiguity between the accuracy of the prediction capability of the algorithm and the convergence error bound. The authors are encourage to adopt a different naming convention, such as error bound, for ..\epsilon..

Table 1 is not referenced in the text.

---

> ### Author Rebuttal · Authors · 2026-03-30
>
> Thank you for the encouraging review and for the detailed suggestions on exposition and experiments.
> ### **Strength And Weaknesses**
> We agree that the presentation would benefit from a clearer introduction of Asynchronous SGD prior to Algorithms 1 and 2. In the revision, we will add a short intuitive description of the asynchronous setup in Section 3, including the role of stale gradients, the meaning of the delay variables, and the distinction between the actual iterate sequence and the virtual sequence used in the analysis.
>
> We also agree that the notation in Algorithm 1 / Eq. (2) can be made clearer. In particular, we will revise the notation around the clipped gradients and the virtual sequence so that it is more transparent which gradients are computed from actual models and how they enter the virtual sequence. We will also make sure that the text around Eq. (2) explicitly explains the relationship between the two sequences and why clipping yields the bound in Lemma 4.1.
>
> Thank you as well for the suggestions on terminology and organization. We will replace “accuracy” by “error bound” when referring to the tolerance $\varepsilon$, and we will make sure to reference Table 1 explicitly in the text.
>
> Finally, you are correct that the current experiments are restricted to the homogeneous simulated shared-memory setting. We have added heterogeneous experiments with a different model architecture, which are shown below. We also plan to release the code upon acceptance to support reproducibility.
>
> ### **New experimental results**
> We have run the CIFAR10 experiments but with a label skew data split. We distribute the data among the workers by sampling $p_k \sim \text{Dirichlet}(0.5)$ and allocating a portion $p_{k,i}$ of the samples in class $k$ to client $i$. Moreover, we use a CNN instead of a ResNet architecture. In Tables 1 and 2 we show simulated wall-clock time to reach 75% test accuracy for delay factors $D \in \{4,8\}$. In order of best to worst performance in these experiments, we have Clipped ASGD ($c=1$), Ringleader ASGD [1], and Vanilla ASGD. Moreover, we will include experiments on the federated version of the Shakespeare dataset in the paper as well.
>
> **Table 3.** Simulated wall-clock time to reach 75% test accuracy with delay factor $D=4$. The maximum wall-clock time is 17,000 time units. The average time per oracle call was 0.337 time units.
>
> | Step size | Vanilla | Ringleader | Clipped c = 1.0 | Clipped c = 2.0 | Clipped c = 4.0 |
> |-|-|-|-|-|-|
> | 0.000977  | 12109| 17000| 17000| 17000| 17000|
> | 0.001953  | 7146| 9819| 17000| 17000| 11900|
> | 0.003906  | 4439| 5757| 17000| 12125| 5741|
> | 0.007812  | 2986| 4076| 10796| 5036| 3077|
> | 0.015625  | 2662| 3383| 4736| 3757| 2313|
> | 0.031250  | 3038| 2369| 2693| 2010| 3054|
> | 0.062500  | 17000| 5789| **2009**| 3120| 6470|
> | 0.125000  | 17000| 17000| 2279| 16779| 17000|
> | 0.250000  | 17000| 17000| 17000| 17000| 17000|
> | 0.500000  | 17000| 17000| 17000| 17000| 17000|
>
> **Table 4.** Simulated wall-clock time to reach 75% test accuracy with delay factor $D=8$. The maximum wall-clock time is 30,000 units. The average time per oracle call was 0.668 time units.
>
> | Step size | Vanilla | Ringleader | Clipped = 1.0 | Clipped = 2.0 | Clipped = 4.0 |
> |-|-|-|-|-|-|
> | 0.000977  | 22415| 30000| 30000| 30000| 30000|
> | 0.001953  | 14034| 20183| 30000| 30000| 24522|
> | 0.003906  | 8788| 10752| 30000| 24915| 12605|
> | 0.007812  | 5989| 8003| 25947| 11197| 6705|
> | 0.015625  | 5250| 5253| 9878| 5864| 4617|
> | 0.031250  | 4614| 4571| 6611| 4568| 4685|
> | 0.062500  | 17767| 18245| **3914**| 4667| 10453|
> | 0.125000  | 30000| 30000| 4606| 30000| 30000|
> | 0.250000  | 30000| 30000| 30000| 30000| 30000|
> | 0.500000  | 30000   | 30000      | 30000         | 30000         | 30000         |
>
> [1] Artavazd Maranjyan and Peter Richt´arik. "Ringleader ASGD: The first Asynchronous SGD with optimal time complexity under data heterogeneity". arXiv preprint arXiv:2509.22860, 2025

---

> > ### Author Rebuttal · Reviewer_W8P9 · 2026-03-31
> >
> > Thank you for taking the time to address my comments and run additional experiments.

---

> > > ### Author Response · Authors · 2026-04-05
> > >
> > > Thank you again for your thorough reading of our paper and the constructive feedback.
> > >
> > > Since the discussion phase does not allow us to upload a revised manuscript, we want to confirm clearly that we will incorporate the changes we committed to in the rebuttal. In response to the points you have raised, we will in the camera-ready version:
> > > 1. include heterogeneous experiments on label-skew CIFAR10 and federated Shakespeare,
> > > 2. improve clarity of notation, change terminology of $\varepsilon$ to "error bound", and reference Table 1 explicitly in the main text,
> > > 3. include a link to a public repo containing the code to run all experiments.

---

### Official Review · Reviewer_dY21 · 2026-03-13

**Soundness:** 3
**Presentation:** 3
**Significance:** 3
**Originality:** 3
**Overall Recommendation:** 5
**Confidence:** 3

**Summary:**

In this paper, the authors consider clipped Asynchronous Stochastic Gradient Descent (ASGD). They provide the first high-probability convergence guarantees for this method under a sub-Weibull noise assumption.

**Compliance With Llm Reviewing Policy:**

Affirmed.

**Final Justification:**

In my initial review, I noted that the paper was well-motivated and presented a theoretical step forward for Asynchronous SGD. However, I raised concerns regarding a lack of contextualization with the broader heavy-tailed noise literature (bounded central moments), missing time complexity analyses, and outdated empirical baselines.

The authors provided a clear rebuttal that addressed these points. First, they clarified their theoretical positioning, acknowledging the broader literature while isolating their contribution to the asynchronous/heterogeneous domain. Second, they provided the requested time complexity bounds and ran new empirical evaluations against the suggested current baselines (Ringmaster and Ringleader ASGD).

By providing the missing mathematical bounds and updating the experiments, the soundness and significance of the paper are strengthened. Assuming these updates are integrated into the final manuscript, I have raised my score and recommend this paper for acceptance.

**Key Questions For Authors:**

No Questions

**Limitations:**

No limitations

**Strengths And Weaknesses:**

### **Strengths**

1. **Presentation & Structure:** The paper is well-motivated and logically structured.
2. **Theoretical Contribution:** The manuscript provides the first high-probability convergence analysis for Asynchronous SGD, which is a notable and interesting theoretical step forward.

### **Weaknesses**

**1. Contextualization of Heavy-Tailed Noise Literature:** The authors rely on a sub-Weibull noise assumption, motivated by the challenge of heavy-tailed noise. However, they omit several critical works that have already established high-probability bounds under significantly more general assumptions. For instance, [1] provides the first high-probability bounds for clipped-SGD and clipped-SSTM under a standard bounded variance assumption, which is strictly more general than sub-Weibull. Furthermore, [2] analyzed clipped-SGD under the $p$-Bounded Central Moment assumption across strongly convex, convex, and non-convex regimes prior to the work of Nguyen et al. (2023). This line of research was later extended to Distributed clipped SGD [3]. Additionally, the authors should contextualize their work against recent high-probability analyses for Normalized SGD [4] and standard SGD operating under the $p$-Bounded Central Moment assumption [5].

**2. Missing Time Complexity Analysis & Outdated ASGD Baselines:** The manuscript skips a substantial body of literature regarding the time complexity of Asynchronous SGD [6-9]. Crucially, the authors do not compute the time complexity for their proposed clipped ASGD method. Furthermore, the paper claims that delay-adaptive SGD is the current state-of-the-art. This claim is outdated. Recent works have proposed Ringmaster ASGD [7] and Ringleader ASGD [8], which achieve optimal time complexity in homogeneous and heterogeneous settings, respectively. To meet the standard of publication, the authors should compute the time complexity of their method and appropriately compare it against these current optimal ASGD baselines.

**References:**

* [1] Gorbunov, E., Danilova, M. and Gasnikov, A., 2020. Stochastic optimization with heavy-tailed noise via accelerated gradient clipping. *Advances in Neural Information Processing Systems*, 33, pp.15042-15053.
* [2] Sadiev, A., Danilova, M., Gorbunov, E., Horváth, S., Gidel, G., Dvurechensky, P., Gasnikov, A. and Richtárik, P., 2023. High-probability bounds for stochastic optimization and variational inequalities: the case of unbounded variance. In *International conference on machine learning* (pp. 29563-29648). PMLR.
* [3] Gorbunov, E., Sadiev, A., Danilova, M., Horváth, S., Gidel, G., Dvurechensky, P., Gasnikov, A. and Richtárik, P., 2023. High-probability convergence for composite and distributed stochastic minimization and variational inequalities with heavy-tailed noise. *arXiv preprint arXiv:2310.01860*.
* [4] Hübler, F., Fatkhullin, I. and He, N., 2024. From gradient clipping to normalization for heavy tailed sgd. *arXiv preprint arXiv:2410.13849*.
* [5] Fatkhullin, I., Hübler, F. and Lan, G., 2025. Can SGD Handle Heavy-Tailed Noise?. *arXiv preprint arXiv:2508.04860*.
* [6] Tyurin, A., Pozzi, M., Ilin, I. and Richtárik, P., 2024. Shadowheart SGD: Distributed asynchronous SGD with optimal time complexity under arbitrary computation and communication heterogeneity. *Advances in Neural Information Processing Systems*, 37, pp.3717-3780.
* [7] Maranjyan, A., Tyurin, A. and Richtárik, P., 2025. Ringmaster ASGD: The first Asynchronous SGD with optimal time complexity. *arXiv preprint arXiv:2501.16168*.
* [8] Maranjyan, A. and Richtárik, P., 2025. Ringleader ASGD: The first Asynchronous SGD with optimal time complexity under data heterogeneity. *arXiv preprint arXiv:2509.22860*.

---

> ### Author Rebuttal · Authors · 2026-03-30
>
> Thank you for the thoughtful review and for pointing out several relevant references.
> ### **Strengths**
> Thank you for acknowledging the contributions of the paper. We would also like to highlight another important contributions in providing a theoretical explanation of previously unexplained behavior in real large scale asynchronous deep learning training (see the third paragraph of Section 1).
> ### **Weaknesses**
> **1. Contextualization of Heavy-Tailed Noise Literature:** We agree that the paper should better contextualize itself within the broader literature on Clipped SGD and high-probability stochastic optimization under heavy-tailed or unbounded-variance assumptions, and we will add the references you listed. It is true that bounded central moment assumptions are broader than the sub-Weibull assumption when comparing noise models in isolation. Our intended claim is not that sub-Weibull is the most general assumption available, but rather that it is a useful heavy-tail model that still allows us to obtain sharp high-probability results in the asynchronous setting. We will revise the text to make this distinction explicit.
>
> Our contribution is therefore complementary to the serial Clipped SGD literature: the novelty here is the asynchronous setting, including the heterogeneous case, together with high-probability guarantees under asynchrony. We believe that these results are novel and non-trivial. We will revise the related-work section to make this comparison more precise and to avoid underrepresenting prior work on Clipped SGD, Normalized SGD, and SGD under bounded central moments.
>
> **2. Missing Time Complexity Analysis & Outdated ASGD Baselines:** We also thank you for pointing out the newer ASGD time-complexity literature. You are right that the current framing is incomplete, and we will update it. In particular, we will remove wording that suggests Delay-adaptive ASGD is the overall state of the art, and instead compare our results more carefully against more recent optimal-time-complexity ASGD methods. We have also included new experimental results with Ringmaster ASGD in the response to Reviewer Viw8, and Ringleader ASGD in the response to Reviewer W8P9. At the same time, we will clarify that the aim of the paper is primarily to isolate and explain the effect of clipping in asynchronous optimization, rather than to propose a new method that is uniformly optimal in time complexity.
>
> Regarding time complexity, we agree this deserves explicit discussion. Since clipping does not change the computation dynamics, ignoring the small overhead of clipping the gradient, the time complexity analysis becomes very similar to normal ASGD. In the homogeneous case under a fixed computation model where workers take $s_1 \leq \dots \leq s_n$ seconds per oracle call. Then, multiplying the oracle complexities with $\left(\sum_{i=1}^n \frac{1}{s_i}\right)^{-1}$ yields the time complexity in the case of full concurrency. If $s_i$ are known, the concurrency and the workers can be chosen so that the time complexity becomes$$\widetilde{O} \left( \min_{\tau_C} \left(\sum_{i=1}^{\tau_C} \frac{1}{s_i}\right)^{-1} \left(\frac{\sigma^2}{\epsilon^4} + \frac{\sigma \tau_C}{\epsilon^3} + \frac{\tau_C}{\epsilon^2} \right)\right).$$ This similarly applies to the high probability guarantee, since the worst case computation dynamics are deterministic under this model.
>
> Quantifying the time complexity is more delicate in the heterogeneous case than the homogeneous one. For this reason, we will include experiments on wall-clock time in the heterogeneous case. In the new heterogeneous experiments, the slowdown of heterogeneous ASGD in time per oracle call was 3x and 6x for delay factors $D=4,8$, respectively. However, in the special case where the average time per oracle call $\bar s$ is uniform over workers but computation times may vary over time, the average time per oracle call of the algorithm is $\frac{\bar s}{n}$ (Lemma 21 of [2]). This case is common when training on data centers with distributed memory.
>
> [9] Chen, J., Monga, R., Bengio, S., and Jozefowicz, R. Revisiting distributed synchronous SGD. In International Conference on Learning Representations Workshop Track, 2016
>
> [10] Mishchenko, K., Bach, F., Even, M., and Woodworth, B. E. Asynchronous SGD beats minibatch SGD under arbitrary delays. Advances in Neural Information Processing Systems, 35, 2022
>
> [11] Koloskova, A., Stich, S. U., and Jaggi, M. Sharper convergence guarantees for asynchronous SGD for distributed and federated learning. Advances in Neural Information Processing Systems, 35, 2022
>
> [12] Wang, Y., Cao, Y., Wu, J., Chen, R., & Chen, J. Tackling the data heterogeneity in asynchronous federated learning with cached update calibration. ICLR 2026
>
> [13] Toghani, M. T., & Uribe, C. A. Unbounded gradients in federated learning with buffered asynchronous aggregation. In 2022 58th Annual Allerton Conference on Communication, Control, and Computing

---

> > ### Author Rebuttal · Reviewer_dY21 · 2026-04-04
> >
> > Thank you for the detailed rebuttal. I appreciate the effort in addressing my concerns.
> >
> > __Contextualization:__ The clarification regarding the _sub-Weibull_ assumption is clear. Framing the contribution as complementary to the serial _Clipped SGD_ literature by extending it to asynchronous and heterogeneous settings is an appropriate positioning of the work. I look forward to seeing the revised related work section with the broader bounded central moment literature included.
> >
> > __Time Complexity & Baselines:__ Thank you for providing the time complexity formulation for the homogeneous case. Additionally, including new experimental baselines against _Ringmaster_ and _Ringleader ASGD_ addresses my concerns regarding the previous baseline comparisons.
> >
> > Given that the missing literature, time complexity analysis, and modern baselines have been addressed, my main concerns are resolved. I will raise my score accordingly.

---

> > > ### Author Response · Authors · 2026-04-05
> > >
> > > Thank you again for your thorough reading of our paper and the constructive feedback. We also thank you raising your score.
> > >
> > > Since the discussion phase does not allow us to upload a revised manuscript, we want to confirm clearly that we will incorporate the changes we committed to in the rebuttal. In response to the points you have raised, we will in the camera-ready version:
> > > 1. clarify the positioning of the paper as complementary to the serial Clipped SGD and heavy-tailed noise literature by extending to the asynchronous and heterogeneous setting,
> > > 2. include the references on Clipped and Normalized SGD, and the bounded central $p$-moment assumption,
> > > 3. include an explicit discussion on the time-complexity of Clipped ASGD, and compare with the recent time-optimal methods Ringmaster and Ringleader ASGD [7,8],
> > > 4. include Ringmaster and Ringleader ASGD in the experiments on wall-clock time to reach target test accuracy.

---

### Official Review · Reviewer_xpoL · 2026-03-13

**Soundness:** 3
**Presentation:** 3
**Significance:** 2
**Originality:** 3
**Overall Recommendation:** 5
**Confidence:** 2

**Summary:**

This paper shows that Clipped ASGD is robust to stragglers under a gradient noise model that captures the heavy-tailed noise commonly observed in deep learning. The authors prove that, in both homogeneous and heterogeneous settings, gradient clipping leads to convergence rates that are independent of the maximum delay. Moreover, the paper provides the first high-probability convergence guarantee for an asynchronous optimization algorithm. Finally, the experimental results corroborate the theoretical findings and confirm the robustness of the proposed method.

**Compliance With Llm Reviewing Policy:**

Affirmed.

**Final Justification:**

Given the authors’ response and their commitment to making the suggested revisions, I am raising my score. I expect these improvements to be reflected in the final version of the paper.

**Key Questions For Authors:**

See the weakness section

**Limitations:**

yes

**Strengths And Weaknesses:**

## Strenghs


1) The paper theoretically establishes a phenomenon that had previously only been observed empirically: gradient clipping stabilizes the asynchronous training of deep learning models. More specifically, in the homogeneous setting, the authors show that their method achieves guarantees that are independent of the maximum delay, matching the performance of prior approaches. In the heterogeneous setting, the paper is the first to provide such a guarantee.

2) The paper is clear and easy to follow.

3) The mathematical derivations are presented clearly and in sufficient detail.

## Weaknessess

1) The authors acknowledge that the heterogeneous version of ClippedASGD may increase the wall-clock time per average oracle call compared to standard ASGD. However, this slowdown is neither quantified theoretically nor evaluated empirically. Could the authors provide bounds on the time complexity of their heterogeneous variant of ClippedASGD and compare them with those of standard ASGD? It would also strengthen the paper to include experiments that empirically measure this slowdown.

2) The technical details underlying the high-probability bounds and the heterogeneous setting, which constitute the two main novelties of the paper, are not discussed in much depth in the main text. Could the authors include a brief proof sketch of Theorems 4.3 and 5.1 to give the reader a clearer and more accessible overview of the main ideas behind the proofs?

---

> ### Author Rebuttal · Authors · 2026-03-30
>
> Thank you for the positive assessment and for highlighting both the theoretical novelty and the clarity of the presentation.
> ### **Weaknesses**
> The heterogeneous version does indeed increase the wall-clock time per oracle compared to homogeneous version due to the worker sampling scheme. But to clarify, standard ASGD does not converge in the heterogeneous case [1]. In order to guarantee convergence we use the same sampling scheme as the heterogeneous version of ASGD in [2] and asynchronous FL algorithms proposed in e.g. [3,4]. That is, the heterogeneous version of Clipped ASGD has the same time per oracle call as these alternatives.
>
> In the homogeneous case we can consider the fixed computation model where workers take $s_1 \leq \dots \leq s_n$ seconds per oracle call [5]. Then, multiplying the oracle complexities with $\left(\sum_{i=1}^n \frac{1}{s_i}\right)^{-1}$ yields the time complexity in the case of full concurrency (i.e., using all workers all the time). If $s_i$ are known, the concurrency and the workers can be chosen so that the time complexity becomes $$\widetilde{O} \left( \min_{\tau_C} \left(\sum_{i=1}^{\tau_C} \frac{1}{s_i}\right)^{-1} \left(\frac{\sigma^2}{\epsilon^4} + \frac{\sigma \tau_C}{\epsilon^3} + \frac{\tau_C}{\epsilon^2} \right)\right).$$ This similarly applies to the high probability guarantee, since the worst case computation dynamics are deterministic under this model.
>
> Quantifying the time complexity is more delicate in the heterogeneous case. For this reason, we will include experiments on wall-clock time in the heterogeneous case. We have included new experimental results in the response to Reviewer W8P9, where the slowdown per oracle call was 3x and 6x for delay factors $D=4$ and $D=8$ compared to the homogeneous versions of ASGD. However, in the special case where the average time per oracle call $\bar s$ is uniform over workers but computation times may vary over time, the average time per oracle call of the algorithm is $\frac{\bar s}{n}$ (Lemma 21 of [2]). This case is common when training on data centers with distributed memory.
>
> We also appreciate the request for more intuition behind the main technical novelties. We will add a short proof sketch of Theorems 4.3 and 5.1 in the main text. At a high level, Theorem 4.3 combines the perturbed-iterate viewpoint with the control obtained from clipping and the sub-Weibull tail bound, which yields a polylogarithmic dependence on the failure probability. For Theorem 5.1, the key point is that clipping controls the perturbation induced by stale heterogeneous updates without introducing a bias towards fast workers which e.g. Delay-adaptive ASGD does.
>
> [1] Mishchenko, K., Bach, F., Even, M., and Woodworth, B. E. Asynchronous SGD beats minibatch SGD under arbitrary delays. Advances in Neural Information Processing Systems, 35, 2022
>
> [2] Koloskova, A., Stich, S. U., and Jaggi, M. Sharper convergence guarantees for asynchronous SGD for distributed and federated learning. In Advances in Neural Information Processing Systems, 35, 2022
>
> [3] Wang, Y., Cao, Y., Wu, J., Chen, R., & Chen, J. Tackling the data heterogeneity in asynchronous federated learning with cached update calibration. In ICLR 2026
>
> [4] Toghani, M. T., & Uribe, C. A. Unbounded gradients in federated learning with buffered asynchronous aggregation. In 2022 58th Annual Allerton Conference on Communication, Control, and Computing
>
> [5] Artavazd Maranjyan and Peter Richt´arik. Ringleader ASGD: The first Asynchronous SGD with optimal time complexity under data heterogeneity. arXiv preprint arXiv:2509.22860, 2025

---

> > ### Author Rebuttal · Reviewer_xpoL · 2026-04-03
> >
> > I thank the authors for their detailed and convincing rebuttal. I also share Reviewer Viw8’s position regarding the various revisions promised by the authors. I am willing to raise my score and support acceptance of the paper, conditional on these promised changes being included in the camera-ready version.

---

> > > ### Author Response · Authors · 2026-04-05
> > >
> > > Thank you again for the careful reading and constructive feedback. We are grateful for you willingness to support acceptance.
> > >
> > > Since the discussion phase does not allow us to upload a revised manuscript, we want to confirm clearly that we will incorporate the changes we committed to in the rebuttal. In response to the points you have raised, we will in the camera-ready version include:
> > > 1. an explicit discussion of the time complexity of the homogeneous and heterogeneous versions of Clipped ASGD,
> > > 2. experiments on the wall-clock time in the heterogeneous setting, empirically showing the slowdown,
> > > 3. a brief proof sketch of Theorems 4.3 and 5.1 to provide an accessible overview of the novel theoretical contributions.

---

### Decision · Program_Chairs · 2026-04-30

**Decision:**

Accept (regular)

**Comment:**

This work addressed asychrony in the case of distributed SGD, in particular, they propose that clipping improves robustness to stragglers.

The reviewers raised several concerns which the authors did their best to address, reviewers increased their scores and expect that the final version of the paper integrates the promised edits